# A genome-wide assessment of the ancestral neural crest gene regulatory network

Dorit Hockman[1,2], Vanessa Chong-Morrison[1,6], Stephen A. Green[3], Daria Gavriouchkina[1,7], Ivan Candido-Ferreira[1], Irving T.C. Ling[1], Ruth M. Williams[1], Chris T. Amemiya[4], Jeramiah J. Smith [5], Marianne E. Bronner[3] & Tatjana Sauka-Spengler [1]*

The neural crest (NC) is an embryonic cell population that contributes to key vertebrate-specific features including the craniofacial skeleton and peripheral nervous system. Here we examine the transcriptional and epigenomic profiles of NC cells in the sea lamprey, in order to gain insight into the ancestral state of the NC gene regulatory network (GRN). Transcriptome analyses identify clusters of co-regulated genes during NC specification and migration that show high conservation across vertebrates but also identify transcription factors (TFs) and cell-adhesion molecules not previously implicated in NC migration. ATAC-seq analysis uncovers an ensemble of *cis*-regulatory elements, including enhancers of *Tfap2B, SoxE1* and *Hox-α*2 validated in the embryo. Cross-species deployment of lamprey elements identifies the deep conservation of lamprey *SoxE1* enhancer activity, mediating homologous expression in jawed vertebrates. Our data provide insight into the core GRN elements conserved to the base of the vertebrates and expose others that are unique to lampreys.

[1] Radcliffe Department of Medicine, MRC Weatherall Institute of Molecular Medicine, University of Oxford, Oxford, UK. [2] Division of Cell Biology, Department of Human Biology, Neuroscience Institute, Faculty of Health Sciences, University of Cape Town, Cape Town, South Africa. [3] Division of Biology and Biological Engineering, California Institute of Technology, Pasadena, CA, USA. [4] Molecular Cell Biology, School of Natural Sciences, University of California, Merced, CA, USA. [5] Department of Biology, University of Kentucky, Lexington, KY, USA. [6]Present address: Division of Biosciences, Faculty of Life Sciences, University College London, London, UK. [7]Present address: Okinawa Institute of Science and Technology, Molecular Genetics Unit, Onna, Japan. *email: tatjana.sauka-spengler@imm.ox.ac.uk

The neural crest (NC) is a migratory embryonic cell population that is unique to vertebrates. NC cells form in association with the developing central nervous system, which they delaminate from after undergoing an epithelial-to-mesenchymal transition (EMT). They subsequently migrate throughout the body to give rise to a plethora of derivative cell types[1]. The advent of the NC with its contributions to numerous tissues and organs is thought to have played an essential role in the diversification of vertebrates[2, 3]. Elucidating how the genetic signals involved in NC specification were modified over the course of vertebrate evolution is key to understanding how this diverse assemblage evolved and expanded[4]. This requires a detailed picture of how the NC GRN functioned in the vertebrate ancestor. To this end, the sea lamprey, a basal vertebrate, serves as a good model. Morphologically, these animals are considered living fossils with a body-plan that has remained consistent over at least the last 400 million years[5].

The current view of the NC GRN has been compiled from data generated in jawed vertebrates[6]. By taking a candidate gene approach to compare lamprey and gnathostome TFs and signalling molecules, we previously showed that many key NC genes were conserved in expression and function between lamprey and jawed vertebrates[7]. These results suggested that the basic NC GRN was already present at the base of vertebrates, although some key regulators were missing from the lamprey NC specifier module[7].

Recently, our understanding of the NC GRN in gnathostomes has been increased with the advent of next-generation sequencing techniques including RNA-seq, ChIP-seq and ATAC-seq[8–12]. Progress in reconstructing the NC GRN of jawless vertebrates, however, has been limited owing to incomplete genomic information. Recently, a germline genome assembly for the sea lamprey, that unlike previous assemblies[13] is not affected by DNA-elimination[14] and has an increased contiguity to near chromosome-scale resolution[15], has made it possible to interrogate the regulatory genome of this basal vertebrate with increased confidence.

Here, we explore the dynamics of gene expression and chromatin accessibility during cranial NC specification and migration in the sea lamprey. By comparing our genome-wide representation of the active lamprey NC transcriptome to that of jawed vertebrates, our analyses highlight the components of the NC GRN that are conserved and likely to be essential for NC specification. We analyse the chromatin accessibility in the NC cells of two lamprey species, and find that cross-species mapping highlights putative *cis*-regulatory elements. Importantly, we identify enhancer elements that drive expression in the lamprey NC, and provide evidence that regulation of a *SoxE* family gene is conserved between jawless and jawed vertebrates. By adapting high-throughput tools to the lamprey, our data provide insight into the ancestral state of the NC GRN.

## Results

**Dynamics of the developing NC transcriptome**. We obtained cranial NC RNA-seq data by dissecting the dorsal neural tube (DNT) including premigratory, early-delaminating and/or late-delaminating NC cells at Tahara (T) stage[16] T18, T20 and T21 (Fig. 1a), respectively. In sea lamprey embryos, NC cells reside within the neural folds, which converge at T18 to form a neural rod and fuse at T20, when the first signs of NC migration have been reported[16,17].

Reads were mapped to the sea lamprey germline genome assembly. A consensus transcriptome consisting of 120,207 transcripts at 72,171 genetic loci was assembled de novo from the mapped DNT data sets, combined with mapped RNA-seq

data sets from whole heads and whole embryos at T20. 67,736 of the transcripts did not overlap with any annotated genes and thus represent candidate novel transcripts or transcribed transposable elements. The latter were not integrated in the current conservative gene model annotation that excluded repetitive elements[15]. Principal component analysis (PCA) of DNT count data showed clear separation along principal component 1 (PC1), which accounted for 90% of the variance, reflecting the developmental stage of the tissue (Fig. 1b). PCA and regression analysis confirmed that the replicate data sets at each stage were highly correlated, demonstrating high reproducibility (Supplementary Fig. 1). Differential expression analysis between the T18 and T21 samples, which represent the neural tube tissue and associated premigratory and late-delaminating cranial NC, respectively, revealed 9106 differentially expressed genes (DESeq2, adjusted *p* value < 0.05). Of these, 5400 were enriched at T21, whereas 3706 were depleted (Fig. 1c). As expected, fewer genes were recovered as differentially expressed when T18 and T20 samples, or T20 and T21 samples were compared (Supplementary Fig. 2a).

We assessed the dynamics of signalling molecules and TFs expressed during NC specification making use of the germline genome annotation in which lamprey gene models were assigned to likely vertebrate homologues[15]. As expected, several bona fide NC markers were enriched at T21 when compared with T18 (Fig. 1c, Supplementary Data 1). *Wnt1*, which has a role in establishing the neural plate border and is maintained in the DNT[18], was one of the most significantly enriched genes, as were *Wnt3* and *Wnt10* (Fig. 1c). In contrast, several *Wnt* homologues (*Wnt5a/b*, *Wnt7a*, *Wnt8a*) were depleted at T21, consistent with studies showing that *Wnt* expression is dynamically modulated during NC delamination and migration[19,20]. Genes play a role in neural tube development, such as *Pax6a* and *BMP4*, as well as NC specifier genes like *SoxE* genes (*SoxE1* and *SoxE2*), *Foxd3*, *Msx2*, *Tfap2A* and *Tfap2B* were increased by at least twofold at T21 when compared with T18, whereas other genes including several *Hox*, *Tbx* and *Gata* TFs were depleted (Fig. 1c), analogous to previous observations in gnathostomes[8]. Similarly, premigratory NC genes (e.g., *SoxE1* and *FoxD3*) were upregulated at T20 when compared with T18 (Supplementary Fig. 2a; Supplementary Data 1). When T21 was compared with T20, late NC specification factors and factors expressed at the onset of migration (e.g., *SoxE2*, *SoxE3*, *CadN*) were enriched (Supplementary Fig. 2a; Supplementary Data 1). Both *Ets1b* and *Twist1* were depleted at T21 when compared with T18 (Fig. 1c), confirming previous findings regarding their absence from lamprey migratory NC[21].

Weighted Gene Co-expression Network Analysis (WGCNA)[22] revealed 12 gene clusters with significantly higher gene expression at T18, and 13 gene clusters with significantly higher gene expression at T21, mirroring the results from our differential expression analysis (Supplementary Data 2, Supplementary Fig. 3). This approach delineated patterns of all genes expressed in NC cells. *Tbx6* and *Wnt5a* were placed in a cluster of 767 genes that showed a drop in expression from T18 to T20, and remained low at T21 (Fig. 1d; Supplementary Data 2: cluster 1). The largest cluster (3193 genes) showed an increase in expression from T18 to T20, maintained at T21. This contained key NC specification module genes[21,23] such as *SoxE1*, *Foxd3*, *Wnt1*, *Pax3/7*, *Msx2* and *Tfap2A* (Fig. 1e, g; Supplementary Data 2: cluster2). Interestingly, these TFs were co-expressed with cell adhesion and cytoskeletal factors involved in NC emigration (Integrin[ITG]A2/A10/B3, Galectin-3 [Lgals3], Interleukin[IL]17, etc.; Supplementary Data 2: cluster2). NC migration module genes, including *SoxE3*, *Tfap2B* and *Gdf7*, were placed in the next largest cluster (1395 genes), which displayed low expression at both T18 and T20 that increased at T21 (Fig. 1f, g; Supplementary Data 2: cluster3).

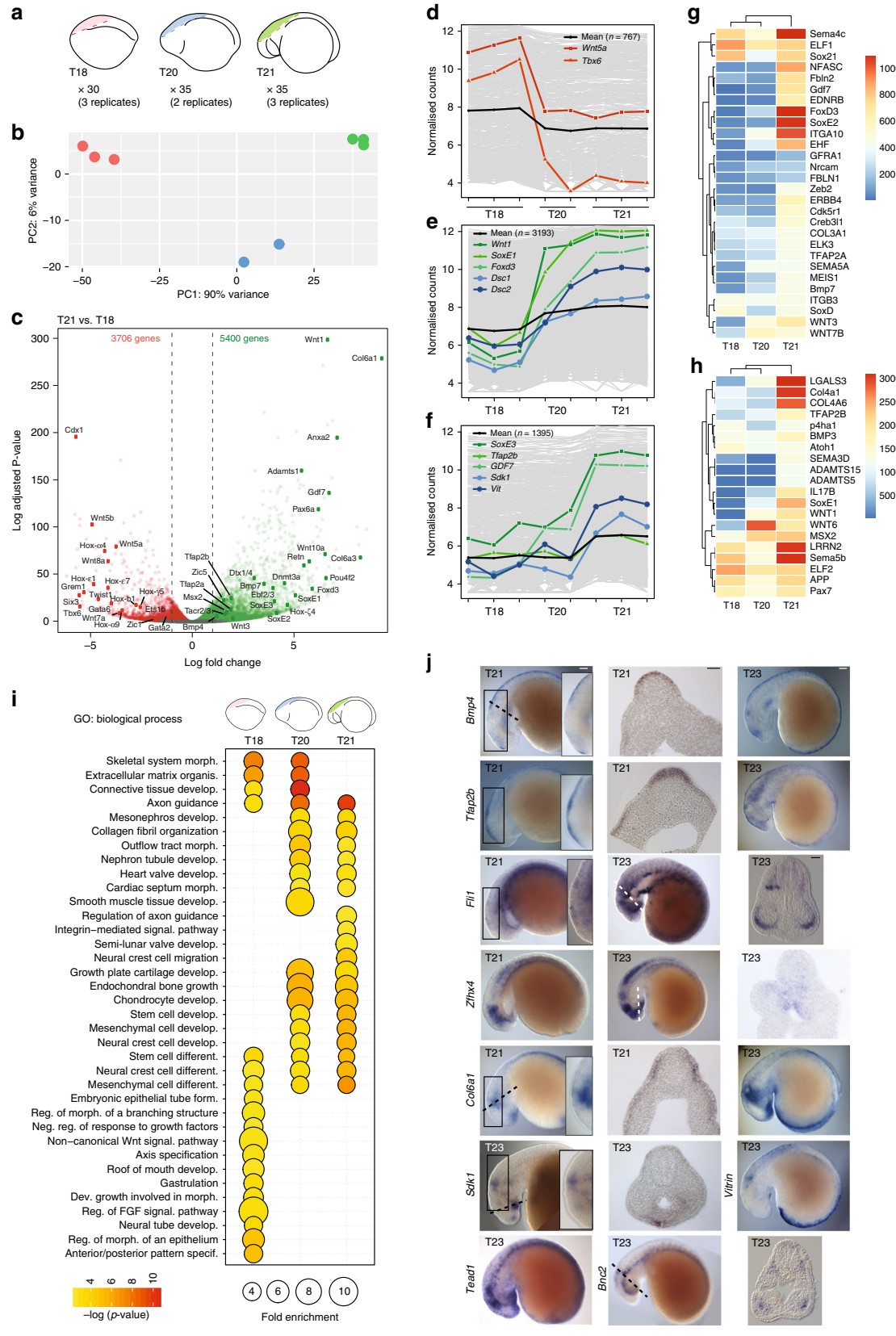

Other putatively co-regulated TFs involved in NC migration were also placed in this cluster (*Sox21* and *Zeb2*), as well as signalling receptors and ligands (*ERBB4*, *Ednrb*, *Sema3D/4 C/5B*), secreted matrix remodelling enzymes (*MMP13*, *ADAM10*, *ADAMTS1*), collagens (*Col3a1/4a1/4a6*) and lamprey orthologues involved in organisation of the extracellular matrix (*Prolyl 4-hydroxylase subunit alpha-1* [*P4ha1*], *Fibulin* [*Fbln2*] and *Creb3l1*) (Fig. 1g; Supplementary Data 2: cluster3). This cluster also featured

**Fig. 1** Dynamics of the developing NC gene expression profile. **a** Schematic depicting the region dissected from T18, T20 and T21 lamprey embryos for DNT RNA-seq and the number of biologically independent samples analysed. **b** PCA of rlog-transformed gene expression count tables for 56,319 genes with non-zero read counts. PC1, which accounts for 90% of the variance is stage dependent (colours indicate stage as in **a**. **c** Volcano plot of differential expression analysis between T21 and T18 ($p$ value < 0.05; green, enriched; red depleted at T21). Coloured dots and labels indicate genes previously known to be enriched or depleted in the developing NC. Dashed line indicates logFoldChange = $1/-1$. **d**–**f** Clusters of highly correlated genes (grey lines) identified by WGCNA (**d**, downregulated after T18; **e**, upregulated at T20; **f**, upregulated at T21; black line is the mean profile), showing specific genes that are known to be downregulated (red) or upregulated (green) in the NC, as well as upregulated genes that have not been previously implicated in NC development (blue). **g**–**h** Heatmaps of the average variance stabilised normalised gene counts for selected genes from WGCNA clusters 2 and 3, showing increased expression at T21. Low-level (**g**) and high-level (**h**) expressing genes are shown. **i** Bubble plots summarising enrichment and $p$ values for the most significant GO biological process terms associated with enriched genes at T18 relative to T21 and at T20 and T21 relative to T18 (only terms enriched more than three-fold are shown). **j** Whole-mount in situ hybridisation for the indicated genes at T21 and T23 (expression patterns observed in at least 3 embryos). Insets are magnifications of boxed regions. Dashed lines indicate approximate plane of sections in the adjacent panel. Scale bars in row 1 and row 3 are the same for images at equivalent stages. Scale bars for wholemount embryo images: 100 μm. Scale bars for sections: 50 μm

downstream effectors ensuring proper differentiation into NC derivatives, such as melanocytes (*RAB32*, *Sox21*), neurons (*Nrcam*, *Atoh1*, *Neurofascin* orthologues, *LRRN2*) and glia (*SoxD*, *GFRA1*, *Cdk5r1*, *APP* orthologues) (Fig. 1g; Supplementary Data 2: cluster3).

Importantly, the two largest WGCNA clusters contained genes that have not previously been implicated in NC development, including genes coding for cell-adhesion molecules, such as desmocolins (*Dsc1/2*) and *Sdk1*, and extracellular matrix proteins, such as *Vitrin* (Fig. 1e, f; Supplementary Data 2). Many novel TFs, as well as those that play a role later in NC development, were also in these clusters (Fig. 1g; Supplementary Data 2). For example, *Nmi* (*N-myc interactor*), which interacts directly with Sox10[24] and inhibits canonical Wnt signalling in cancer[25], showed increased expression at T20, whereas *EHF* (*Ets homologous factor*, also known as *Epithelial Specific Ets-3*), proposed to have a role as a tumour-suppressor in prostate cancer[26] and oncogene in ovarian cancer[27], showed elevated expression at T21. *Fli1* and *Satb2*, which are expressed in the developing branchial arch cartilage and mesenchyme[28,29], and *Nfatc*, which forms a complex with Sox10 during Schwann cell differentiation[30], were also elevated at T21 (Supplementary Data 2).

Gene Ontology (GO) analysis of genes enriched at T18, and thus depleted at T21, highlighted terms associated with the NC GRN signalling module, including Wnt and FGF signalling (Fig. 1i). Terms associated with early embryonic processes such as gastrulation, axial patterning and neural tube development were also enriched at this stage. Conversely, genes enriched at T20 or T21 when compared with T18 revealed an overrepresentation of terms associated with early NC specification, and the development of tissues that receive a NC contribution, such as cartilage, bone, neural tissue and the heart (Fig. 1i; Supplementary Fig. 2b). The biological process terms associated with NC cell migration were specifically enriched only at T21. Although the GO term 'NC cell differentiation' was enriched at all stages (Fig. 1i), the genes associated with this term at NC stages, T21 and T20 showed little overlap with those at T18 (Supplementary Fig. 2c; Supplementary Data 3), thus providing evidence for a dynamic progression in gene expression as NC development progresses. The presence of GO terms associated with kidney development at T20 and T21 is largely owing to enrichment in genes known to be highly expressed in the NC, but also have a role elsewhere in the embryo (e.g., *Ctnnb1*, *Sox9* and *Bmp4*).

Finally, we characterised the expression of a number of genes that were enriched at T21 using whole-mount in situ hybridisation (Fig. 1j; Supplementary Fig. 2d). As expected, *Bmp4* and *Tfap2b* were clearly expressed in the DNT at T21, whereas *Tfap2b* was maintained in the migratory NC at T23. Interestingly, *Fli1* expression was also associated with the DNT at T21, indicating that this factor may have an early role in NC development, as well

as being expressed in the branchial arches later on. Similarly, *Zfhx4*, which is enriched in the migratory NC in the chicken[12] was associated with the neural tube and branchial arches, suggesting a role in both early and later NC development. *Col6A1* was most strongly expressed adjacent to the neural tube, surrounding the developing otic placode, where it likely supports NC cell migration[31]. *Sdk1* and *Vitrin* were expressed in the region coinciding with the neural tube and delaminating NC at T23. *Sdk1* was also associated with the developing branchial arch mesenchyme at T23. *Tead1*, a TF that is involved in regulating cell proliferation during development and in cancer[32], was expressed in the neural tube as well as the migratory NC at T23. The zinc finger protein *Bnc2*, which is implicated in craniofacial development[33] and adult pigmentation[34], was associated with the DNT and the developing branchial arch mesenchyme at T23. *Sdk1*, *Vitrin* and *Bnc2* expression at T21 was possibly too low for detection by in situ hybridisation (normalised readcount of 50.8, 119.4 and 378.4, respectively), whereas *Tead1* was expressed most strongly in the mid-brain region including the NC (Supplementary Fig. 2d).

Taken together, our RNA-seq data confirm, with a higher level of detail, previous findings that a large proportion of the NC GRN is conserved to the base of the vertebrates[2,21]. Importantly, our analyses reveal factors, whose role in NC development and diversification warrants further investigation.

**Genome-wide assessment of chromatin accessibility.** We next sought to explore the regulatory connections between players in the NC GRN. To this end, it is essential to identify *cis*-regulatory elements that control gene expression in the NC. ATAC-seq reveals regions of accessible chromatin, and enables a genome-wide assessment of putative *cis*-regulatory elements[35]. We analysed chromatin accessibility in lamprey cranial DNTs or whole heads at T20, T21 and T23 (Fig. 2a), which encompass the early-delaminating, late-delaminating and post-migratory NC. Mapped biological replicate ATAC-seq libraries were highly correlated (Supplementary Fig. 4) and insert size distribution showed a stereotypical ~ 150 bp periodicity (Supplementary Fig. 5a), consistent with the nucleosome occupancy of chromatin[35].

Peak detection and annotation, using our de novo assembled transcriptome, was consistent across stages, with the majority of peaks found in intergenic and intronic regions where *cis*-regulatory elements are expected to be located (Fig. 2b). To focus our analyses on peaks associated with NC GRN genes, consensus peaksets for each annotation category (promoter, intergenic and intronic) were filtered to contain peaks that were associated with genes enriched at T21. Promoter peak groups were further filtered to contain elements associated with the promoters of genes annotated in the germline genome assembly.

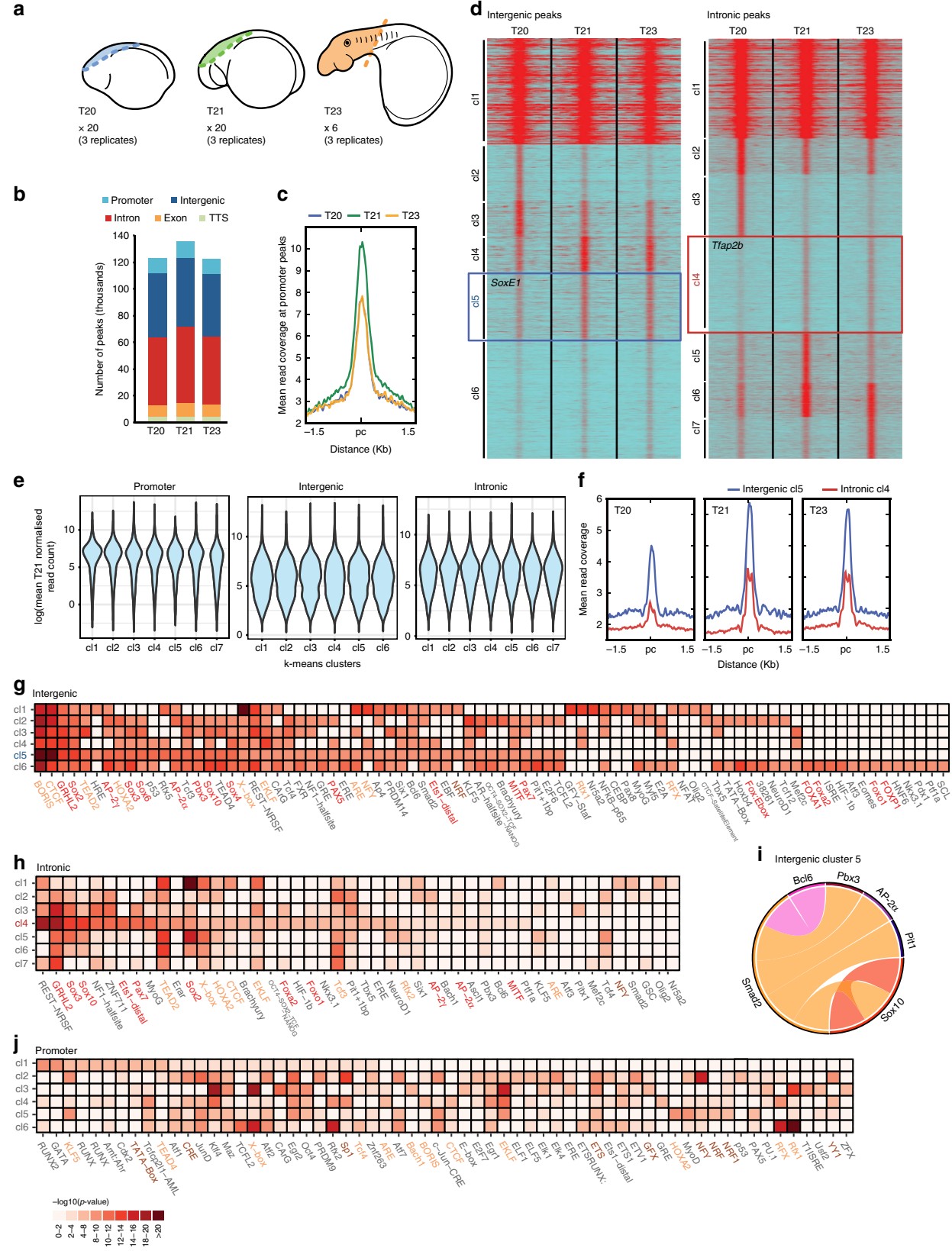

*K*-means clustering of the ATAC-seq signal over consensus peaksets (8998 intergenic; 17,908 intronic; 1860 promoter) revealed the dynamics of chromatin accessibility genome-wide over the course of development (Fig. 2c–f). The ATAC-seq signal associated with promoter peaks was highest at T21 for all clusters showing, as expected, that enriched gene expression correlated with increased promoter accessibility (Fig. 2c).In addition, when all promoter peaks were taken into consideration (i.e., 10,286 annotated and novel promoter peaks), gene expression associated with promoter peaks was higher and less variable than that

**Fig. 2** Profiling of chromatin dynamics in the developing NC. **a** Schematics indicating the region of DNT or head dissected from T20, T21 and T23 lamprey embryos for ATAC-seq and the number of biologically independent samples analysed. **b** Genomic functional annotation of our ATAC-seq peaksets for all stages. **c** Mean ATAC-seq read coverage map at each stage over our consensus promoter peakset (i.e. peaks associated with T21 enriched genes), showing higher read coverage at T21. **d** Heatmaps depicting *k*-means linear enrichment clustering of ATAC-seq reads at all stages across consensus intergenic and intronic peaksets. Boxes indicated the large "EMT" clusters that show enriched signal at T21 and T23. **e** Violin plots visualising the distribution of mean normalised T21 read counts for genes associated with *k*-means clusters. Gene expression associated with promoter peak clusters (annotated and novel promoters) is higher and less variable than that for genes associated with intergenic and intronic clusters. **f** Mean ATAC-seq read coverage maps at each stage for "EMT" clusters (intergenic cluster 5 in blue; intronic cluster 4 in red), showing higher coverage at T21 and T23. **g**, **h**, **j** TF-binding motif enrichment analysis for intergenic (**g**), intronic (**h**) and promoter (annotated and novel) (**j**), *k*-means clusters. NC master regulator motifs are highlighted in red. Similar motifs shared between intergenic and intronic cluster 1 and promoter clusters are highlighted in orange. Canonical promoter motifs are highlighted in brown. **i** TFs that were significantly enriched in pair-wise in silico co-binding analyses conducted on intergenic *k*-means cluster 5. cl, cluster; Pc, peak centre

associated with the intergenic and intronic peak clusters (Fig. 2e, Supplementary Fig. 5b).

We were particularly interested *cis*-regulatory elements that regulate gene expression during EMT. *K*-means clustering of intergenic and intronic peaks revealed two large clusters (intergenic cluster 5; intronic cluster 4) that displayed increased accessibility at T21 and T23 compared with T20 (Fig. 2d, f). Gene ontology terms associated with intergenic cluster 5 included 'regulation of localisation', 'positive regulation of cell-substrate adhesion' and 'regulation of cell motility', whereas terms associated with intronic cluster 4 included 'cell–cell junction organisation', 'cytoskeleton reorganisation' and 'positive regulation of cell migration' (Supplementary Fig. 5c). In addition, these clusters contained elements associated with known NC GRN TFs, *SoxE1* (intergenic cluster 5) and *Tfap2B* (intronic cluster 4).

To quantify the significance of these EMT clusters, we plotted the ATAC-seq signal levels of our peaksets at T20 against those at T23 and calculated the Pearson correlation coefficient for all intergenic and intronic clusters (Supplementary Fig. 5d). Both EMT clusters were significantly offset from all other identified groups of accessible elements, suggesting that the dynamic opening of putative *cis*-regulatory elements may single them out as functional enhancers during.

We used TF-binding site motif analysis to further interrogate the ATAC-seq *k*-means clusters. Similar to our recent ATAC-seq analyses in zebrafish[11] and chicken[12] NC, several intergenic clusters, including cluster 5, were enriched for Sox, Tfap2 and Tead motifs (Fig. 2g, h). GRHL2, Pax, Ets1 and Fox motifs were also enriched. Intronic cluster 4 displayed similar enrichment for Sox and Pax motifs, as well as Fox sites, whereas GRHL2 and REST-NRSF motifs showed the highest enrichment in this cluster (Fig. 2g, h). The presence of motifs for key NC specification TFs (Sox, Fox, Tfap, Ets1, Pax families[6]), as well those that play a role in EMT (GRHL2[36]), suggested that these clusters harbour *cis*-regulatory elements that provide connections between NC GRN players. Enrichment of CTCF-binding sites in all intergenic clusters further suggests these peaks may represent putative *cis*-regulatory elements.

Cluster 1 for intergenic and intronic peaksets had a distinct TF-binding site profile from the other clusters, which resembled the binding profile of our promoter peakset (annotated and novel promoter peaks), and was enriched for motifs found in the HOMER promoter motif library (Fig. 2j). These clusters consisted of peaks with a broad, open profile at all stages (see Fig. 2d). Therefore, it is likely that cluster 1 for both intergenic and intronic peaksets represent peaks with promoter-like activity.

Combinatorial TF binding at enhancers has been suggested to have an important role in NC GRN function[11]. We used in silico two-way combinatorial analysis to test for evidence of multiple TF co-activity at putative *cis*-regulatory elements. Focussing on intergenic cluster 5, we selected 18 TFs that displayed enriched

motifs and enriched gene expression at T21 (Supplementary Data 4). The motifs of six TFs (Smad2, Sox10, TFAP2a, Pit1, Bcl6 and Pbx3) were significantly enriched in pair-wise analyses (*$p <$ 0.05; two-tailed Chi-squared test) (Fig. 2i). Combinations of Sox10 sites were enriched, whereas Smad2 sites were enriched in combination with all the other five motifs. Interestingly, we found similar co-binding of Smad proteins with canonical NC TFs, such as Sox and TFAP2 proteins in the early chick NC[12]. This suggests that the combinatorial activity of TGFβ-signalling with canonical NC specification factors at enhancers is a conserved property of the NC GRN.

**Active NC-specific *cis*-regulatory elements**. To identify active NC-specific *cis*-regulatory elements, we used peaks from our EMT clusters that were associated with loci of known NC GRN genes in enhancer-reporter assays. An element, located 16.6 kb downstream of the *SoxE1* locus (Fig. 3a), drove specific reporter expression in the delaminating NC cells from T21 and labelled the cells as they migrated into the branchial arches and contributed to known NC-derived structures, such as the branchial arch cartilage (Fig. 3b). Analysis of reporter expression alongside endogenous *SoxE1* expression revealed an overlap in the delaminating NC at T23 and in the branchial arch cartilage at T26 (Fig. 3c). Together, these results suggest that we have identified a lamprey enhancer for *SoxE1*, which is activated in the delaminating NC. An element located in the third intron of the *Tfap2B* gene drove reporter expression in the migrating NC from T23 and labelled NC derivatives at later stages (Fig. 3d, e).

These *cis*-regulatory elements overlapped with peaks in similar ATAC-seq data sets from the brook lamprey (*Lampetra planeri*) (Fig. 3a, d). This surprising finding suggests conservation of *cis*-regulatory elements across lamprey species, which were separated at least 40 MYA[37]. Our analysis thus suggests a high degree of sequence conservation at the level of the functional non-coding regions of the genome of these two species, thus facilitating the identification of *cis*-regulatory elements using cross-species whole-genome alignment of ATAC-seq data.

**Putative lncRNAs associated with lamprey *Hox-α2* enhancers**. Our ATAC-seq data set can be used to refine known *cis*-regulatory regions into modules that drive expression in specific tissues, including the NC. Sea lamprey *cis*-regulatory elements for *Hox-α2* that drive gene expression in the neural tube, somites and NC are found in a 9-kb region upstream of *Hox-α2*, whereas elements that drive expression in the NC and somites alone are located within 4-kb of the *Hox-α2* locus[38]. Recently, a core 1.5-kb region within the 4-kb enhancer (elementA) was shown to recapitulate the activity of the full 4-kb element[39]. These findings are confirmed by our ATAC-seq data set. We found that a ~ 1.5 kb element encompassing an ATAC-seq positive region at ~ − 8.5 kb drove

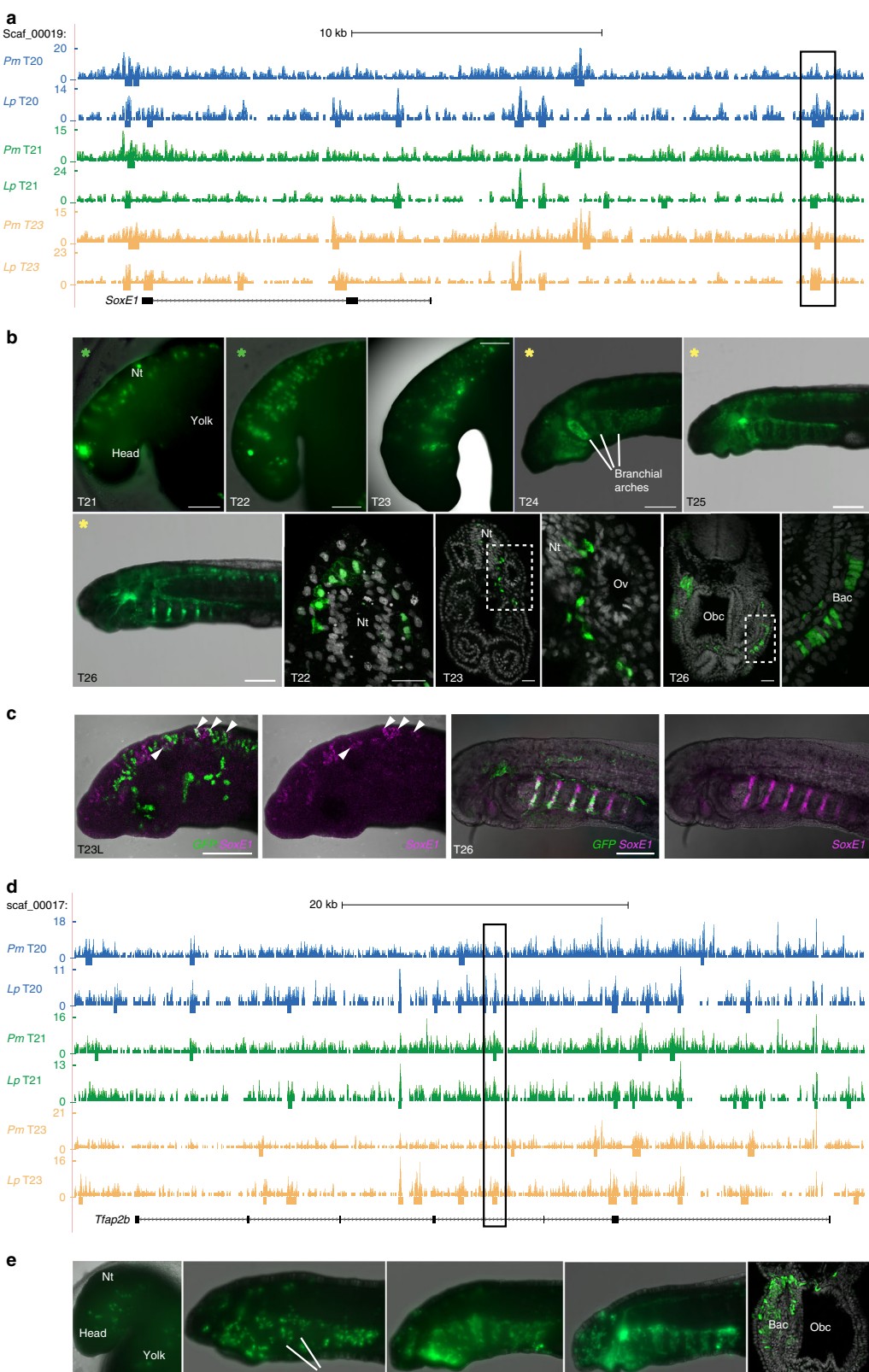

reporter expression that was restricted to the neural tube (Fig. 4a, b: element-8.5 kb). In addition, a clear ATAC-seq-positive region was present over the 1.5 kb core NC enhancer element (Fig. 4a: elementA).

Interestingly, our RNA-seq data revealed bidirectional transcription from these loci, a known occurrence at active

enhancers[40] (Fig. 4a, bottom panel). Two novel transcripts from our transcriptome overlapped these regions: a 12,770 bp sense transcript (Fig. 4a, maroon label) and a 4206 bp, spliced antisense transcript (Fig. 4a, blue label). The longer sense transcript resembles the multiexonic enhancer (me)RNA transcripts reported in association with the ethryroid-specific intergenic

**Fig. 3** Tissue-specific enhancer activity in the lamprey NC. **a, d** The *SoxE1* **a** and *Tfap2B* **d** loci of the sea lamprey germline genome, with merged replicate ATAC-seq coverage tracks from the sea lamprey (*Pm*) and brook lamprey (*Lp*) at each developmental stage. Bars below coverage plots indicate peak regions identified with Macs2. The black box indicates the region tested in enhancer-reporter assays. **b** GFP reporter expression in lamprey embryos injected with the *SoxE1* enhancer-reporter construct at the one-cell stage and allowed to grow to indicated stages (observed in 195 out of 1337 injected embryos). In transverse section (row 2, panels 2–6) GFP+ cells are visible delaminating from the neural tube at T22 ($n = 2$), migrating between the neural tube and otic vesicle at T23 ($n = 2$) and contributing to the branchial arch cartilage at T26 ($n = 3$). Coloured stars indicate panels showing the same embryo at successive developmental stages. Dashed boxed regions indicate regions magnified in adjacent panels. **c** Overlap of *GFP* reporter expression with native *SoxE1* expression (magenta) in the delaminating and migrating NC at T23L (arrowheads; $n = 4$) and in the branchial ach cartilage at T25 ($n = 2$). **e** GFP reporter expression in lamprey embryos injected with the *Tfap2B* enhancer-reporter construct at one-cell stage and allowed to grow to indicated stages (observed in 25 out of 340 injected embryos). In transverse section at T26 (panel 5), GFP+ cells are visible in the branchial arch cartilage ($n = 3$). Bac, branchial arch cartilage; L, late; Nt, neural tube; Obc, orobranchial cavity; Ov, otic vesicle. Scale bars for wholemounts: 200 μm. Scale bars for sections: 25 μm

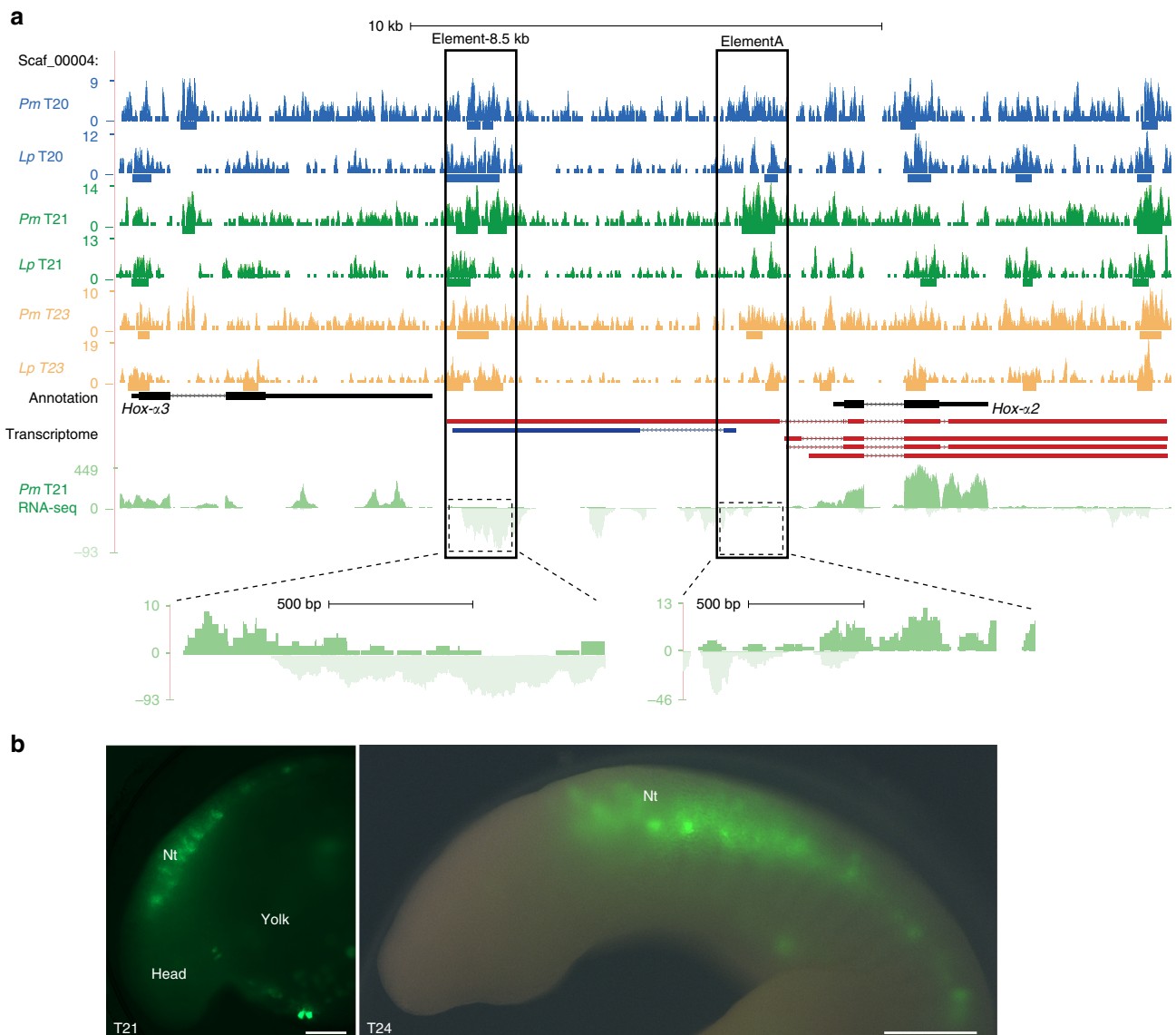

**Fig. 4** Characterisation of a *Hox-α2* enhancer and associated transcription. **a** The *Hox-α3/Hox-α2* locus in the sea lamprey germline genome, with merged replicate ATAC-seq coverage tracks from the sea lamprey (*Pm*) and brook lamprey (*Lp*) at each developmental stage, as well as RNA-seq coverage tracks from a representative T21 sample indicating directional transcription. Bars below ATAC-seq coverage plots indicate peak regions identified with Macs2. The black boxes highlight two ATAC-seq positive regions within the 9-kb region upstream of the *Hox-α2* locus and the dashed boxes highlight bidirectional transcription over these regions. De novo assembled transcripts for the *Hox-α2* locus are shown maroon (sense) and dark blue (antisense). **b** GFP reporter expression in lamprey embryos injected with the element-8.5 kb enhancer-reporter construct at one-cell stage and allowed to grow to indicated stages (observed in 10 out of 149 injected embryos). GFP reporter expression is seen in the neural tube. Scale bars: 200 μm

enhancer, R4 in mice[41], as well as with multiple late NC-specific enhancers in zebrafish[10]. As is the case with the R4 meRNA, the *Hox-α2*–8.5 kb upstream enhancer appears to initiate the transcription of an alternative first exon, which is spliced onto an adjacent annotated exon and reads through the remaining exons of the *Hox-α2* gene. This results in a spliced transcript reminiscent of the annotated version, albeit with an extended 5′-UTR or first exon. Therefore the enhancer, located ~ 8.5 kb upstream of the *Hox-α2* locus, may be acting as alternative promoter (indeed, the peak was annotated as a promoter in our analyses). Alternatively, this transcript might be a byproduct or a facilitator, of chromatin looping linking the upstream enhancer to the *Hox-α2* promoter.

The antisense transcript is one of 6257 putative long non-coding (lnc)RNAs identified in our transcriptome. In all, 48% of these overlap with predicted lncRNA from adult sea lamprey brain, heart, kidney and gonad RNA-seq datasets[15], whereas 70% were associated with ATAC-seq positive regions. The gnathostome *HoxA* locus is known to harbour lncRNAs, including HOTAIRM1[42] and HOTTIP[43], which have been shown to modulate gene expression in *cis*. The putative lncRNA, identified between *Hox-α3* and *Hox-α2*, is significantly enriched in the DNT at T21 when compared to T18 (threefold change; *p.adj.* = 2.9E-49), suggesting it may regulate *Hox-α* expression in the neural tube and/or NC at T21.

**Lamprey *SoxE1* enhancer activity is conserved in gnathostomes**. Our study seeks to define the core components of the NC GRN that are conserved across vertebrates. This includes assessing whether the activity of NC enhancer elements present in a basal jawless vertebrate is conserved in jawed vertebrates, despite 500 million years of independent evolution[37]. We generated transgenic zebrafish carrying the lamprey *SoxE1* enhancer upstream of a minimal promoter and GFP using the Activator (Ac)/Dissociation (Ds) (Ac/Ds) transposition system[44], which facilitates highly efficient transgenesis in zebrafish[45], and resulted in seven independent integrations of the *SoxE1* enhancer:GFP cassette. Although only weak reporter expression was visible in $F_0$ embryos, the $F_1$ generation displayed striking heterospecific reporter expression in the branchial arches by ~ 30 hpf, mirroring the enhancer activity in the lamprey at T23 (Fig. 5a, Supplementary Fig. 6a, b). Later (~ 60 hpf), reporter expression was visible in head structures that receive NC contributions including the branchial arch mesenchyme, cranial ganglia (Fig. 5; Supplementary Fig. 6c–e), as well as in melanocytes, and putative Schwann cells in the trunk (Fig. 5 row 2).

To further investigate the extent to which the lamprey *SoxE1* enhancer activity is conserved among the gnathostomes, we used bilateral neural tube electroporation to introduce a lamprey *SoxE1* enhancer:citrine reporter construct into the developing chicken NC at Hamburger Hamilton stage(HH)8. At HH18, citrine expression was visible in the branchial arches and in the dorsal root ganglia, alongside and occasionally overlapping NC2:cherry reporter expression, which was used a marker for the NC[46] (Fig. 5b).

The observed lamprey *SoxE1* enhancer activity in gnathostome NC suggests that the TF-binding code characterised by our genome-wide motif analysis (Fig. 2g) and present in the lamprey *SoxE1* enhancer can be recognised by gnathostome TFs despite a lack of sequence conservation within the whole region that would identify a homologous regulatory element. To test this, we searched for significantly enriched TF-binding motifs in the *SoxE1* enhancer. We found binding sites for key NC TFs, including SoxE, Smad and Tfap2 factors, as well as a Hox site, which would be expected to restrict the enhancer activity in the

cranial region to the hindbrain NC streams (Supplementary Fig. 7; HOMER log-odds scores: SoxE: 8.35; Smad: 6.42; Tfap: 8.77; Hox: 9.08). To assess the conservation of these putative binding sites we compared the sea lamprey genome sequence in the region of the *SoxE1* enhancer to the same region in the juvenile brook lamprey and our brook lamprey ATAC-seq data (Supplementary Fig. 7). The majority of the binding sites were conserved across species, with only point mutations in two putative SoxE sites and one putative Hox-binding site (Supplementary Fig. 7). The high level of sequence conservation in this region supports a functional role for this enhancer across lamprey species and the presence of conserved TF-binding motifs suggests these sites may have a role in mediating cross-species enhancer activity.

To test whether cross-species enhancer activity is mediated by combinatorial TF activity in vivo, we used CRISPR/Cas9 to knockout the endogenous expression of selected TFs (*hoxa2b*, *hoxb2a*, *hoxb3a*, *tfap2a*, *sox10*) in zebrafish embryos, resulting from crossing our lamprey *SoxE1* enhancer transgenic line reporting EGFP to a transgenic line expressing DsRed in the branchial arches (Fig. 5d–e). When DsRed-positive cells were sorted from the surrounding tissue in experimental embryos (Cas9 injected with target guide RNAs) and control embryos (Cas9-only injected), we found a significant downregulation of EGFP expression in the branchial arch cells upon targeted TF knockout as compared with controls (Fig. 5d, e; Supplementary Fig. 8). These perturbations support the notion that one or a combination of these short TF-binding site motifs is able to convey enhancer function[47] and mediate the regulatory activity across evolutionary time, despite the lack of extensive sequence conservation.

Overall, our analyses indicate that the regulation of *SoxE* gene expression in the migratory NC is conserved to the base of the vertebrates and that enhancers with such conserved activity reflect the central lynchpin mediating the conservation of the NC GRN. Importantly, this suggests that regulation of one of the central players in the NC GRN has remained constant as the existence of the last common ancestor between jawed and jawless vertebrates.

## Discussion

We present the most complete assembly to date of the lamprey NC GRN that can be directly compared with that of gnathostomes. Our data enable identification of tissue-specific enhancers whose activity is evolutionary conserved and reveals putative non-coding RNA species.

Analysis of the lamprey NC transcriptional network provides global insight into the evolution of NC transcriptional programmes. At premigratory NC stages (T20) in the lamprey, we observed similar gene enrichments in categories equivalent to those observed in zebrafish[11] and chicken[12]. The lamprey premigratory NC shows significant functional enrichment in categories associated with mesenchymal (smooth muscle, connective tissue, cartilage and bone development) and neuronal (axonogenesis, gliogenesis) NC derivative fates. As development progresses and bona fide NC cells begin to delaminate (T21), enrichment terms changed to those characterising NC and stem cell programmes as well as autonomic nervous system formation. Interestingly, the lamprey NC GRN lamprey differs from that of zebrafish in that it lacks the NC sub-programme involved in the specification of the enteric nervous system (ENS). This is consistent with studies showing that the lamprey may lack vagal NC and that the lamprey ENS may have much later onset[48].

Analysis of co-expression clusters using WGCNA increases the resolution of the putative lamprey NC GRN[21] and suggests links

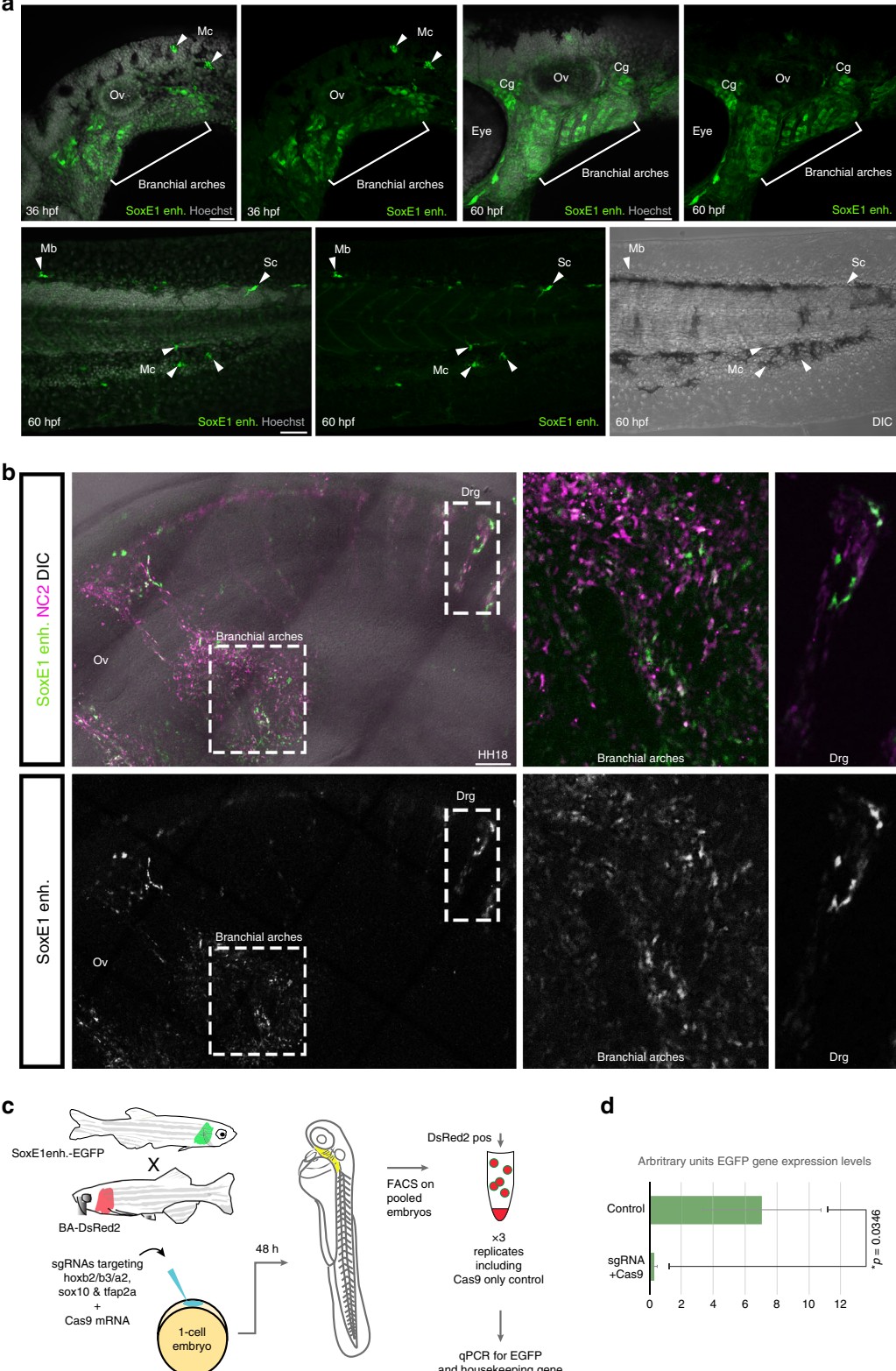

**Fig. 5** The activity of the lamprey *SoxE1* enhancer is conserved in gnathostomes. **a** In a 36 hpf transgenic zebrafish GFP reporter expression is visible in the developing branchial arches and melanocytes (*n* = 3). At 60 hpf, GFP+ cells populate the branchial arch cartilage and cranial ganglia in the head, whereas GFP+ melanocytes, melanoblasts and a putative Schwann cell are visible in the trunk (row 2) (*n* = 6). **b** In a HH18 chicken embryo, GFP expression is present in the branchial arches and the dorsal root ganglia (Drg) alongside cherry expression driven by the chick *Foxd3* NC2 enhancer (*n* = 15). Dashed boxed regions indicate regions magnified in adjacent panels. **c** Schematic of CRISPR/Cas9 TF knockout experiment in transgenic zebrafish. **d** qPCR results indicating the change in EGFP expression levels relative to bactin in the branchial arches after injection with either Cas9 mRNA alone or Cas9 mRNA together with sgRNAs against *hoxb2a*, *hoxb3a*, *hoxa2b*, *sox10* and *tfap2a*. Error bars: SD. Cg, cranial ganglia; Drg, dorsal root ganglia; Mb, melanoblasts; Mc, melanocytes; Ov, otic vesicle; Sc, putative Schwann cell. Scale bars in (**a**) 50 μm. Scale bar in (**b**) 200 μm

between early specification factors and their downstream effectors. TFs and signalling molecules associated with the NC specification module (e.g., *Pax3/7*, *FoxD3*, *Tfap2A*, *SoxE1*) are upregulated at T20 and co-regulated with genes associated with delamination, including cell adhesion and cytoskeletal factors. Later, at T21, TFs associated with NC migration (e.g., *Sox21*, *Zeb2*, *Tfap2B*) are co-expressed with genes associated with active migration, including signalling receptors, their ligands and secreted matrix remodelling enzymes. Interestingly, these genes are co-expressed with genes involved in NC differentiation. Future analysis using this resource to investigate direct links between co-regulated genes will allow expansion of the global structure of the lamprey NC GRN.

Our analysis of chromatin dynamics in the lamprey NC enabled the identification of tissue-specific enhancers. Similar to studies from *C. elegans*, where ATAC-seq analysis of whole animals at early embryonic, larval and adult stages revealed dynamically regulated *cis*-regulatory regions with tissue-specific activity[49], we have performed this analysis using dissected cell populations predominantly, but not exclusively, comprised of NC cells at three successive stages of lamprey development. Together, these studies show that the dynamic signature associated with changes in chromatin accessibility over time can be used to pinpoint putative tissue-specific regulatory regions.

Interestingly, we show that heterospecific analysis of ATAC-seq data from different lamprey species can provide clues for the location of conserved *cis*-regulatory regions. Although the evolutionary distance between the sea lamprey and gnathostomes precludes identification of *cis*-regulatory elements based on sequence conservation, the ATAC-seq reads from brook lamprey, *L.planeri*, were successfully mapped cross-species to the genome of the sea lamprey, *P. marinus*. This suggests that the putative functional non-coding elements have been conserved between the two lamprey species over the last 40 million years[37]. Thus, mapping the brook lamprey ATAC-seq data to the sea lamprey genome has enabled identification of conserved genomic regions.

We show that a lamprey *SoxE* enhancer drives tissue-specific reporter expression in the zebrafish and chicken NC. Interestingly, experiments in the invertebrate chordate, amphioxus, where the entire amphioxus *SoxE* locus and flanking genes were integrated into the zebrafish genome, resulted in reporter expression in the neural tube and tail bud, but not in the NC[50]. This suggests that a NC enhancer for *SoxE* expression is not present in the vicinity of the *SoxE* locus in amphioxus. Together, these results support the hypothesis that the acquisition of novel enhancers in early vertebrates was critical for the evolution of the NC-specific gene expression. Gain-of-function *cis*-regulatory changes, such as the appearance of new TF binding sites, likely facilitated co-option of pre-existing gene batteries, including the pro-chondrocytic *SoxE* genes and other mesenchymal gene programmes, into NC-like cells at the neural plate border[2]. Indeed, we show that the lamprey *SoxE1* enhancer harbours putative binding site motifs for important NC TFs, including Tfap2 and SoxE factors, that activate and maintain *SoxE* transcription in the chick[51,52], zebrafish[53] and lamprey[21]. A Hox site is also present and could possibly control the activity pattern of the enhancer confined to specific regions of cranial NC conserved across vertebrate taxa[38]. By knocking out the TFs associated with these binding sites in our transgenic zebrafish carrying the lamprey enhancer, we provide evidence to support the hypothesis that either single or a combination of TF binding sites mediate the conserved enhancer activity in the NC.

Taken together, our results suggest that the evolution of key TF-binding motifs was central to NC GRN evolution and that conservation of TF binding at these sites is important for the conservation of the NC GRN. Previous analyses of cis-regulatory

TF-binding patterns across gnathostomes showed that, whereas binding motifs are conserved across species, their alignment or positioning is not[54]. This suggests that the functionality of *cis*-regulatory regions is independent of sequence constraint. We provide functional evidence that this hypothesis holds true to the base of the vertebrates by revealing that a *cis*-regulatory sequence from an agnathan, that does not align to the gnathostome genome, is able to drive tissue-specific gene expression in two gnathostome species. This conserved regulatory grammar may have been maintained over evolutionary time through positive-selection promoted by a requirement for co-operative binding of diverse TFs to direct tissue-specific gene expression. Support for this hypothesis comes from a recent study that analysed the location of binding sites for four liver-essential TFs across five mammalian species and found that *cis*-regulatory modules that harbour sites for all four TFs were more likely to be conserved than those with fewer shared sites[55]. It is plausible that physical interactions between the TFs have a role in the evolution and selection of higher order combinatorial TF binding. Indeed, thermodynamic biophysical modelling shows that the evolution of a functional binding site can be accelerated by cooperativity between adjacent TFs[56].

In summary, by taking advantage of our highly contiguous germline genome assembly[14], we have presented a genome-wide representation of gene expression and chromatin dynamics during lamprey cranial NC development. A limitation of our approach, which relied on dissection, is that analysed samples included some dorsal neural and ectodermal tissues, and thus our data represent a mixed cell population rather than a pure NC one. However, by focussing our analyses on the genes and chromatin regions that are being dynamically modified during the analysed time-points we were able to extract the expected signature for the cranial NC cells, which unlike skin and neural cells within this mixed cell population, are highly metabolically active and transcriptionally dynamic, owing to a later onset of their specification in the embryo. Taken together, our analyses uncover critical components of the NC GRN that are shared across vertebrates, as well as expose new players whose further investigation will expand our current view of the genetics of NC development.

## Methods

**Lamprey husbandry and embryo dissections**. Adult sea lamprey (*Petromyzon marinus*) were supplied by the US Fish and Wildlife Service and Department of the Interior. Embryos obtained by in vitro fertilisation, were grown to the desired stage in compliance with California Institute of Technology Institutional Animal Care and Use Committee protocol #1436. Brook lamprey (*L. planeri*) embryos and ammocoete larvae were collected from a shallow river in the New Forest National Park, United Kingdom, with permission from the Forestry Commission and maintained in filtered river water at 13–19 °C. Prior to dissection, embryos were dechorionated in 0.1 × Marc's Modified Ringers buffer (MMR) in a dish lined with 1% agarose. For dissection, embryos were moved into a shallow well in the agarose. T18, T20 and T21 DNTs including premigratory, early-delaminating and/or late-delaminating NC cells were dissected from the head using an eye-lash knife. T20 and T23 heads were dissected using forceps.

**RNA extraction and library preparation**. RNA was extracted from groups of at least 30 dissected DNTs at each stage, as well as from whole heads (two groups of 20) and whole embryos (two groups of 10) at T20. Tissue was lysed in the Ambion RNAqueous Total RNA Isolation kit lysis buffer (AM1931), set on ice for 15 mins with occasional vortexing, flash frozen in liquid nitrogen and stored at − 80 °C. RNA was extracted using the Ambion RNAqueous Micro Total RNA isolation kit and assessed using the Agilent Bioanalyser. Sequencing libraries were prepared from 100 ng RNA per sample using the NEBNext Ultra Directional RNA Library Prep Kit for Illumina (E7420) in combination with the NEBNext Poly(A) mRNA Magnetic Isolation Module (E7490) and NEBNext High-Fidelity 2 × PCR Master Mix (M0451S). Libraries were indexed and enriched by 15 cycles of amplification. Library preparation was assessed using the Agilent TapeStation and libraries quantified by Qubit. The concentration of library pools was assessed with the KAPA Library Quantification Kit (KK4835). Multiplexed library pools were sequenced using paired-end 75–100 bp runs on the Illumina NextSeq500 platform

for DNT libraries and on the Illumina HiSeq2500 platform for T20 heads and embryos.

**ATAC and library preparation**. Groups of dissected tissue were collected into L-15 medium (Lifetech) with 10% fetal bovine serum at 19 °C. Tissue was first dissociated by pipetting up and down in dispase (1.5 mg/ml in DMEM; 10 mM Hepes, pH7.5), followed by the addition of an equal volume of trypsin (0.05% Trypsin; 0.53 mM EDTA in HBSS) at room temperature for a total of up to 15 mins. Dissociated cells were passed over a 40-μm cell strainer into Hanks' solution (1 × HBSS; 10 mM Hepes; 0.25% BSA) and centrifuged at 500 × g for 7 mins at room temperature. The supernatant was removed and fresh Hank's solution applied. 50,000 cells were counted out and centrifuged for 5 minutes at 500 × g at 4 °C and washed with cold 2/3 phosphate-buffered saline (PBS) by centrifugation for 5 minutes at 500 × g, 4 °C. The cells were resuspended in 50 μl cold lysis buffer (10 mM Tris-HCl, pH7.4; 10 mM NaCl; 3 mM MgCl$_2$; 0.1% Igepal) and centrifuged for 10 mins at 500 × g at 4 °C. Cells were tagmented using the Illumina Nextera kit (FC-121–1030) in a 50 μl reaction for 30 mins at 37 °C. To stop tagmentation, EDTA was added to final concentration of 50 nM and the reaction was incubated at 50 °C for 30 mins. Tagmented DNA was cleaned up using the Qiagen MinElute PCR purification Kit (28004) and amplified using the NEB Q5 High-Fidelity 2 × Master Mix (M0492S) for 14 cycles. The amplified library was cleaned up using the Qiagen MinElute PCR purification Kit (28004) and XP AMPure beads (Beckman Coulter A63880). Tagmentation efficiency was assessed using Agilent TapeStation and libraries quantified by Qubit. The concentration of ATAC library pools was assessed with the KAPA Library Quantification Kit (KK4835). Multiplexed library pools were sequenced using paired-end 40 bp runs on the Illumina NextSeq500 platform. The high correlation of the mapped ATAC-seq signal between biological replicates at each stage (Pearson's $R > 0.9$) confirms the reproducibility of our experimental approach (Supplementary Fig. 5).

**Pre-processing of next-generation sequencing reads**. Read quality was evaluated using FastQC (http://www.bioinformatics.babraham.ac.uk/projects/fastqc/). Reads were trimmed to remove low quality bases using Sickle (https://githubcom/najoshi/sickle) using the parameters -l 30 -q 20.

**RNA-seq analysis**. Reads were mapped to the sea lamprey germline genome assembly[15] using STAR (v2.4.2)[57] (STAR --genomeDir $GENOME --readDatasIn $R1.fastq $R2.fastq --runThreadN 4 --outDataNamePrefix $PREFIX --readDatasCommand zcat --outSAMstrandField intronMotif --alignEndsType EndToEnd --outReadsUnmapped Fastx --outSAMtype BAM SortedByCoordinate). Separate transcriptomes for DNT sample datasets or head and embryo sample data sets were passed over a 40-μm cell strainer Cufflinks followed by Cuffmerge using default parameters[58] to make a consensus transcriptome from all the data sets. Read counts for DNT data sets were obtained with Subread featureCounts (v1.4.6-p4)[59] using the Cuffmerge consensus transcriptome in SAF format as a reference (featureCounts -p -B -M -F SAF -s 2 -T4 -a $SAF -o $OUT $IN.bam). Differential expression and PCA were performed on the DNT readcount data sets using DESeq2 (v.1.8.2)[60]. Weighted correlation network analysis (WGCNA)[22] was performed on the variance stabilised normalised gene count tables generated by DESeq2 (filtered to only contain genes with a normalised count > 8) according to the pipeline detailed in the online WGCNA tutorial (https://labs.genetics.ucla.edu/horvath/CoexpressionNetwork/Rpackages/WGCNA/). A soft-thresholding power of 20 and minimum module size of 100 was used for WGCNA. Both of these analyses were run on the R platform (v3.2.1; http://www.R-project.org/). The average normalised gene counts that were associated with ATAC-seq peakset clusters for stage T21 samples (see ATAC-seq analysis) were plotted in R using ggPlot geom-violin. Output transcript lists from the differential expression analysis and WGCNA were annotated using the gene models associated with the sea lamprey germline genome assembly. *Hox* and *Sox* genes shown in figures were manually annotated with lamprey-specific gene names. Heatmaps of the average variance stabilised normalised gene counts were generated in R using pheatmap. Gene Ontology (GO) analysis was performed on annotated differentially expressed gene sets using the genes with baseMean value > 20, log2 FoldChange > 1 for upregulated and < −1 for downregulated genes and *p*-adjusted value < 0.01. The PANTHER Overrepresentation Test (v11)[61] was used with complete GO term databases for *Mus musculus* (**$p < 0.01$ Binominal test with Bonferroni correction). Output GO terms were filtered to only contain terms that were enriched by at least threefold.

To identify putative lncRNAs in our transcriptome, first transcripts that overlapped with coding genes in the germline genome annotation on the same strand were eliminated using bedtools(v.2.15.0)[62] intersect. The remaining transcripts were used in a blastx search using default parameters against the UniProt/Swiss-Prot database. Any transcripts that shared > 30% sequence identity with known proteins with an e-value > 1E$^{-2}$ were eliminated. Any unspliced transcripts were removed and, using bedtools intersect, the list of putative lncRNAs were limited to transcripts that were within 5 kb of a coding gene and originated from the opposite strand to this closest gene. Subread featureCounts was used to determine the length of the remaining transcripts (featureCounts -p -B -F SAF -s 2 -T4 -a $SAF -o $OUT $IN.bam), and those < 200 bp in length were eliminated.

**ATAC-seq analysis**. Reads were mapped to the sea lamprey germline genome assembly[15] using Bowtie2[63] (bowtie2 --phred33 -p4 -X 2000 --very-sensitive -x $GENOME −1 $R1.fastq −2 $R2.fastq -S $OUT.sam). Duplicates were removed with Picard (v1.83; https://broadinstitute.github.io/picard/) MarkDuplicates and the distribution of fragment sizes assessed with Picard (v1.83) CollectInsertSizeMetrics. Replicate bam files for each developmental stage were merged with SAMtools[64] and filtered with BamTools[65] to remove unpaired reads and reads mapped to the mitochondrial chromosome. Filtered bam files were down-sampled to match the file with the lowest number of reads using Picard (v1.83) DownsampleSam. Downsampled bam files were sorted by name using SAMtools and paired-end bed files were obtained using bedtools(v.2.15.0)[62] bamtobed bedpe. Reads were extended to a read length of 100 bp. Peak-calling was performed using MACS[66] (macs2 callpeak -t $IN.bed −f BED −name $IN.macs2 --outdir $OUT --shiftsize=100 --nomodel --slocal 1000). Output peak files (.xls) for each developmental stage were converted to bed format and merged with bedtools merge to create one consensus peakset. The consensus peakset was annotated with HOMER (v4.7; http://homer.salk.edu/homer/ngs/) annotatePeaks.pl using the Cuffmerge gene models (with genes less than 1500 bp in length removed) as a reference. Annotated peaks were separated into intergenic, intronic and promoter peaksets according to their HOMER annotation. Promoter peaks were filtered with bedtools flank to only include elements that overlapped a region of up to 2 kb upstream of the sea lamprey germline genome gene models (bedtools flank -i promoters.bed -g germline_genome.chrom.sizes -l 2000 -r 0 −s). Intergenic peaks that overlapped with promoters annotated in the sea lamprey germline genome gene models (i.e. gene models that were not present in the de novo cuffmerge assembly) were identified with bedtools intersect and moved to the promoter peakset. The intergenic and intronic peaksets were further filtered to only contain peaks that were <50,000 bp away from genes that were enriched at stage T21 in comparison with T18 (see RNA-seq analysis). *k*-means clustering of ATAC-seq signal over the final peaksets was carried out using SeqMINER[67](1,500 bp on each side; no auto-turning; wiggle step: 15; *k*-means enrichment linear). Read counts for the ATAC-seq signal were obtained with Subread featureCounts (v1.4.6-p4) using the peakset clusters in SAF format as a reference (featureCounts -p -F SAF -T4 -a peaksetCluster.saf). Correlation analysis on ATAC-seq readcount data was performed in R using plot and cor (method="pearson"). GO analysis was performed on the differentially expressed genes associated with intergenic and intronic "EMT" clusters using the PANTHER Overrepresentation Test (v11)[61] with complete GO term databases for *Mus musculus*. Output GO terms were filtered to only contain terms that were enriched by at least 1.8-fold. Remaining GO terms were summarised with REVIGO[68] and subsequently filtered to only contain terms with −log10 *p* value < −1.5.

Motif analysis was performed on the intergenic and intronic peakset clusters with HOMER (v4.7) findMotif.pl using a random set of 2329 (average number of peaks across all intergenic and intronic clusters) non-coding genomic regions as control. Heatmaps were generated in R using ggPlot geom-tile. To assess predicted co-binding frequencies for 18 selected TF motifs (selected based on their expression levels) enriched in intergenic cluster 5 (*SoxE1* enhancer cluster, Supplementary Data 4), all possible pairs of motif combinations were computed in R (v. 3.2.1), using intergenic cluster 6 as a control. HOMER (v.4.7) annotatePeaks.pl script was used to screen such motifs in windows of ±250 bp from the peak centre. Motif instances were then converted into a matrix of presence (1) or absence (0) of motif occurrences in individual genomic regions. This matrix was then used as input into a custom script to calculate motif co-occurrences. Motif co-occurrences enriched at α = 5% (two-tailed Chi-squared test) with Bonferroni correction for multiple hypothesis (m) testing were retained for *P* values < α/m. Co-binding predictions were plotted using the Circlize package in R (v.3.2.3). HOMER (v4.7) annotatePeaks.pl with options −m and −mscore, was used to locate enriched motifs that were identified using findMotif.pl in the full-length *SoxE1* enhancer.

*L. planeri* ATAC-seq data mapped to *SoxE1* enhancer genomic region of the sea lamprey germline genome was extracted using samtools view. The consensus sequence for all stages was generated in IGV (v.2.3.60).

**In vivo enhancer-reporter assays in lamprey**. A region of ~ 1.5 kb encompassing the ATAC-seq positive accessible chromatin region to be tested was amplified by PCR from sea lamprey genomic DNA using primers designed with SnapGene (Clontech), cloned into the HLC GFP reporter vector[69] by In-Fusion HD cloning (Clontech) and sequenced. ISec-I meganuclease-mediated transgenesis[38,69] was performed in sea lamprey embryos. At 2–6 hours post fertilisation, single-celled embryos were injected with the ISec-I vector digestion mix at 20 ng/μl and maintained at 18 °C in 0.1 × MMR for the remainder of their development. At 1 dpf embryos were transferred to 96-well plates until 6 dpf when they were returned to petri dishes, and screened daily for reporter expression. Live embryos were imaged on a depression slide using a Zeiss Scope.A1 microscope fitted with a Zeiss AxioCam MRm camera and Zeiss ZEN 2012 software (blue edition).

**Chromogenic whole-mount in situ hybridisation**. Lamprey embryos were placed in glass vials and fixed overnight at 4 °C in MEMFA. They were then rinsed in PBST, dehydrated in a graded series of ethanol washes and stored in 100% ethanol at − 20 °C. Embryos to be analysed were transferred to fresh 100% ethanol in 1-dram glass vials. Embryos were rehydrated and rinsed three times for 5 mins

each in PBST. Embryos were treated with 20 µg/ml solution of proteinase K in PBST at room temperature, for precisely three minutes, then immediately rinsed twice in PBST. Samples were then washed five minutes in 0.1 M Triethanolamine (TEA) a total of three times. Embryos were treated with 0.25% Acetic Anhydride in TEA for 12 mins, and then in 0.50% Acetic Anhydride for another 12 mins. Embryos were washed 2 × 5 mins in PBST, then refixed in 4% Formaldehyde (diluted from 37% Formaldehyde in PBST) and rinsed 3 × 5 mins or more in PBST. Embryos were then rinsed 3 × 5–10 mins in Hybridisation Buffer (hyb), then incubated for 3 h at 68 °C in 500 µl of fresh hyb. This solution was replaced with fresh, hot hybridisation buffer preincubated with each appropriate antisense digoxigenin-labelled riboprobe diluted to ~ 1 ng/mL of hyb and vials were incubated at 68 °C overnight with slow, minimal shaking. After 12–16 h, hybridisation solution was removed and replaced with fresh hyb. Samples were washed 3 × 30 min in pre-heated hyb solution. Most hyb was removed, except for ~ 500 µL, and an equivalent volume of pre-heated 2 × SSC was added, lightly mixed, and vials were incubated at 68 °C for 15 mins. Samples were then washed 3 × 30 mins in 2 × SSC, followed by 3 × 30 mins in 0.2× SSC. Glass vials were allowed to slowly equilibrate to room temperature, then embryos were rinsed several times in RT MABT, then rinsed into MABT block (1 × MAB plus 2% Roche blocking solution and 0.1% Tween). Embryos were incubated in 500 µL of fresh MABT block for 1 h at RT, and then blocking solution was replaced with antibody solution and left to incubate overnight at 4 °C with slow shaking. Vials were washed twice with MABT block, 30 min each, then washed 5 × 30 mins in MABT, and 2 × 60 mins or more in MABT. Embryos were removed from borosilicate vials and placed into 12-well culture plates, rinsed 3 × 10 min in alkaline phosphatase buffer. Buffer was replaced with BM Purple and samples were kept dark. When colour reaction was complete, embryos were rinsed in MABT, then post-fixed in 4% paraformaldehyde overnight at 4 °C. For imaging, embryos were dehydrated in a graded series of ethanols, rinsed in 100% methanol 4–5 times, and cleared for 3 × 3–5 mins in Murray's Clear (1:2 mix of benzyl alcohol and benzyl benzoate). Murray's Clear was removed and replaced with Permount, and embryos were individually mounted on slides, with coverslips having clay feet to act as spacers. Embryos were visualised on a Zeiss Axioimager.

**Lamprey cryosectioning and immunostaining.** Embryos were fixed at 4 °C overnight in 4% paraformaldehyde in PBS. Fixed embryos were incubated in PBS with 5% sucrose for 4 h at room temperature, followed by incubation overnight at 4 °C in 15% sucrose in PBS. Embryos were transferred into pre-warmed 7.5% gelatine in 15% sucrose in PBS and incubated overnight at 37 °C, before being transferred to pre-warmed 20% gelatine in PBS. Embryos were embedded in rubber moulds and frozen by immersion in liquid nitrogen. Blocks were cryosectioned at 6–10 µm. Gelatine was removed from the slides by a 5-minute incubation in PBS pre-warmed to 37 °C. For Immunostaining, slides were incubated overnight at room temperature in Alexa 488 conjugated anti-GFP antibody (Rabbit, 1:250; Life Technologies; A21311) in blocking solution (10% donkey serum in PBS with 0.1% Triton X-100). Sections were imaged on a Zeiss LSM 780 inverted confocal microscope with Zeiss ZEN 2011 (black edition).

**_L. planeri_ SoxE1 enhancer PCR.** _L. planeri_ genomic DNA was extracted from the heads of three individual ammocoete larvae. Tissue was treated with Proteinase K overnight at 50 °C, followed by phenol-chloroform extraction and ethanol precipitation. PCR was first performed with the same primers used to clone the _P. marinus_ SoxE1 enhancer (5′-tccctcgaggtcgacgaattGCGGTGGCGAGCCGA-3′; 5′-gaggatatcgagctcgaattTGGCGTGGCCAGATCTCG-3′). This product was further amplified in a secondary PCR using primers specific to the _L. planeri_ genomic sequence obtained from ATAC-seq data (5′-GAGTTCGACTTCAGCTCACG3-′; 5′-CCACTCTCATCTCCCAATGAC-3′). PCR products were sequenced and resulting sequences were merged using Fragment Merger[70] and aligned using SnapGene (Clontech) to generate a consensus sequence. The consensus sequence was aligned to the _P. marinus_ SoxE1 enhancer sequence from the germline genome assembly and the _L. planeri_ embryonic ATAC-seq consensus sequence using Clustal Omega in SnapGene (Clontech). The presented alignment was prepared with BOXSHADE (https://embnet.vital-it.ch/software/BOX_form.html) using the "ASCII_differences" output format.

**Zebrafish husbandry and creation of transgenic lines.** This study was carried out in accordance to procedures authorised by the UK Home Office in accordance with UK law (Animals [Scientific Procedures] Act 1986) and the recommendations in the _Guide for the Care and Use of Laboratory Animals_. The lamprey SoxE1 enhancer was cloned into the Ac/Ds-E1b-EGFP vector (http://www.addgene.org/102417/) using In-fusion cloning (Clontech) and co-injected with Ac transposase mRNA into one-cell-stage zebrafish embryos. Injected F_0s were screened for founders. Positive F_1s were grown to reproductive age and backcrossed to F_0s to obtain embryos with bright expression.

**Zebrafish whole-mount immunostaining and cryosectioning.** Zebrafish embryos were fixed in 4% paraformaldehyde for 1 h at room temperature and washed in PBT (1 × PBS containing 0.5% Triton X-100 and 2% DMSO). When necessary embryos were bleached prior to being blocked in either 10% Donkey

serum or 3% BSA in PBT for 2 h and washed in antibody solution [Rabbit anti-GFP, 1:200 in block, Torrey Pines, TP401; mouse (IgG2b) anti-Elavl3/4 (HuC/D), 1:500 in block, Invitrogen, A-21271] overnight at 4 °C. Embryos were washed several times in PBT before adding the secondary antibody (1:200; Alexa 488 donkey anti-rabbit; ThermoFisher Scientific; A21206; Alexa 568 donkey anti-mouse; ThermoFisher Scientific; A10037) in combination with Hoescht (1:1000) for 2 h at room temperature. After several PBT washes, embryos were imaged in whole-mount on a Zeiss LSM 780 upright multiphoton confocal microscope with Zeiss ZEN 2011 (black edition). To obtain sections of whole-mount immunostained samples, embryos were cryoprotected in 30% sucrose in PBS and embedded in OCT by immersion in liquid nitrogen. Sections were imaged on a Zeiss LSM 880 inverted confocal microscope with Zeiss ZEN 2.3 (black edition).

**Fluorescent in situ hybridisation chain reaction.** For in situ hybridisation chain reaction (HCR)[71], a kit containing a DNA probe set, a DNA HCR amplifier, and hybridisation, wash and amplification buffers were purchased from Molecular Instruments for each target mRNA. The _GFP_ and _SoxE1_ probes initiate B3 (Alexa-546) and B4 (Alexa-647) amplifiers, respectively. Embryos were fixed overnight at 4 °C in MEMFA. They were then rinsed in PBST, dehydrated in a graded series of ethanol washes and stored in 100% ethanol at − 20 °C. Following rehydration, embryos were treated with 10 mg/ml Proteinase K for 2.5 min and post-fixed in 4% paraformaldehyde for 20 min. Embryos were incubated with the probes in hybridisation solution overnight at 37 °C and, following appropriate washes, incubated in the hairpin solution in amplification buffer overnight at room temperature protected from light. Embryos were imaged in whole-mount using a Zeiss LSM 780 upright multiphoton confocal microscope with Zeiss ZEN 2011 (black edition).

**Splinkerette PCR.** For splinkerette analysis[72] five positive F2 zebrafish embryos from a single F1 parent outcrossed to wild type were collected and genomic DNA extracted. Approximately 500 ng of genomic DNA was digested overnight with AluI in a 30 µl reaction. Digested genomic DNA was purified using phenol-chloroform followed by ethanol precipitation before ligation with annealed splinkerette adaptors (CGAATCGTAACCGTTCGTACGAGAATTCGTACGAGAATC GCTGTCCTCTCCGGCCACAGGCGATTAT and ATAATCGCCTGTGGCCAA ATCTATACGTATAGAT) using T4 DNA ligase at 16 °C overnight in a thermal cycler. The adaptor-ligated genomic DNA was purified using Zymo Research Clean & Concentrate (Cat. # D4003) and 20 ng of purified product used in a primary PCR reaction. PCR was performed using the following primers: CGAATCGTAACCG TTCGTACGAGAA (binding to adaptor) and GTTTCCGTCCCGCAAGTTAA (binding to Ds-3′ integration arm), with 63 °C annealing temperature and 3 min extension time. 1 µl of primary PCR reaction was then used in 50 µl nested PCR reaction using the following primers: TCGTACGAGAATCGCTGTCCTCTC (binding to adaptor) and CGGTAGAGGTATTTTACCGAC (binding to Ds-3′ integration arm), with 60 °C annealing temperature and 5 mins extension time. The nested PCR was run on agarose gel to visualise number of integrations.

**Chicken embryo in ovo electroporation.** The lamprey SoxE1 enhancer was cloned into the pTK vector[73] with EGFP replaced by citrine, using In-fusion cloning (Clontech). Fertilised wild-type chicken eggs were obtained from Henry Stewart & Co (Norfolk). HH8 chicken embryos were electroporated in ovo. The SoxE1-pTK-citrine and NC2-pTK-mCherry[46] constructs were injected into the neural tube at 3.0 µg/µl each and electroporated bilaterally with 3 + 3 50 ms pulses at 12.5 V with 100 ms rest between pulses. The embryos were incubated at 37 °C until HH18 and imaged in whole-mount using a Zeiss LSM 780 upright multiphoton confocal microscope with Zeiss ZEN 2011 (black edition).

**CRISPR/Cas9 assessment of SoxE1 upstream factors.** Oligo templates for guide RNAs (sgRNAs) targeting splice acceptor/exon boundaries of functional domains for _hoxb2a_, _hoxb3a_, _hoxa2b_, _sox10_ and _tfap2a_ were annealed and RNA in vitro transcribed using HiScribe T7 Quick High Yield RNA Synthesis Kit (New England Biolabs, E2050), followed by purification using MEGAclear Transcription Clean-Up Kit (ThermoFisher, AM1908). To maximise mutagenesis, a pool of up to two sgRNAs per target gene (Supplementary Table 1) were microinjected with Cas9 mRNA[74] into one-cell stage embryos obtained by crossing Tg(SoxE1_dwnstrm1-E1b:EGFP)_ox 164_ and Tg(tcf21:DsRed2)_pd37 75_. For each condition (Cas9 mRNA only, Cas9 mRNA with sgRNAs), ~ 25–30k DsRed2-positive cells that included branchial arches were obtained from 3 days post fertilisation (dpf) embryos by FACS on BD FACSAria Fusion. Total RNA was extracted and DNaseI-treated according to manufacturer's protocol using RNAqueous-Micro Total RNA Isolation Kit (ThermoFisher, AM1931). Reverse transcription was performed on three biological replicates using GoScript Reverse Transcriptase Mix + Oligo(dT) (Promega, A2790). To quantify levels of EGFP transcripts, qPCR was performed using Fast SYBR Green Master Mix (ThermoFisher, 4385612) on Applied Biosystems 7500 Fast system using the following primers: EGFP at 300 nM final concentration: F_EGFP (AAGGGCATCGACTTCAAGGA) and R_EGFP (TGATGCCGTTCTTCTGC TTG); bactin at 150 nM final concentration: beta-actin_E3 (AATCCCAAAGCCAAC AGAGA) and beta-actin_E4 (ACATACATGGCAGGGGTGTT) with three technical replicates per biological replicate. Results were analysed using the delta–delta Ct method.

**Reporting summary**. Further information on research design is available in the Nature Research Reporting Summary linked to this article.

## Data availability

The authors declare that all data supporting the findings of this study are available within the article and its supplementary information files or from the corresponding author upon reasonable request. The data sets generated during and/or analysed during the current study have been deposited in the NCBI GEOarchive database under accession code: GSE112072 https://www.ncbi.nlm.nih.gov/geo/query/acc.cgi?acc=GSE112072.

## Code availability

The custom script used to calculate motif co-occurrences is available at https://github.com/tsslab/chick_NC-GRN. Details of software versions and parameters used, when different from default, are indicated in the Methods section.

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

## Acknowledgements

We thank Sebastian Shimeld for access to embryos and genomic DNA of the brook lamprey, Hugo Parker for the lamprey HLC vector, and Sally-Ann Clark at the WIMM FACS Core Facility for assistance with cell sorting. We thank Toros Tasgin and Sarah Meyes for cloning candidate genes for in situ and Justine Van Greenen for amplifying and analysing *L. planeri* genomic DNA sequences. This work was supported by a Leverhulme Research Grant to T.S.S. (RPG-2015–026), the National Institute of General Medical Sciences of the National Institutes of Health grants to J.J.S. (R01GM104123) and C.T.A. (R24GM095471), a Wellcome Trust Institutional Strategic Support Fund grant (H2RZKC00) to D.H. and T.S.S., a Junior Research Fellowship (Trinity College, Oxford), the Sydney Brenner Fellowship, a Company of Biologists Travelling Fellowship (DEVTF-150403) and an EMBO Short Term Fellowship to D.H., and a Clarendon Fund Fellowship to V.C.M.

## Author contribution

D.H. and T.S.S. conceived this research programme. D.H. generated RNA-seq and ATAC-seq data, performed and analysed lamprey reporter expression assays and performed bioinformatics analysis. V.C.-M. performed zebrafish transgenesis, splinkerette assay, CRIPSR/Cas9 experiments and immunostaining. S.G. performed lamprey whole-mount in situ hybridisation. D.G. assisted in the analysis of RNA-seq and ATAC-seq data. I.C.F assisted in ATAC-seq data analysis. I.L. performed chicken embryo electroporations and imaging. R.W. performed in situ HCR and whole-mount in situ hybridisations. J.S. and C.T.A. provided access to the draft sea lamprey germline genome assembly. M.E.B. provided access to sea lamprey embryos. D.H. and T.S.S. discussed ideas and interpretations and wrote the manuscript. D.H., M.E.B., and T.S.S. edited the manuscript and all authors commented on it. T.S.S. supervised the study.

## Competing Interests

The authors declare no competing interests.
