## [Peer Review File · Nature Communications]

Reviewers' Comments:

Reviewer #1:

Remarks to the Author:

The report describes the use of RNA-seq and ATAC-seq to identify genes and cis-regulatory elements involved in neural crest development in lampreys. Several putative cis regulatory elements are then validated by testing in lamprey and zebrafish. The work is technically impressive and well-executed. Novel features of the work are the choice of model systems, the depth of analysis, and the functional testing of putative elements. The results will serve as a comprehensive resource for neural crest biologists, and those interested in gene regulatory networks and their evolution. My only concern is that impact outside of the neural crest field may be limited. The work supports conservation of the gene network driving neural crest development, and confirms a few previously identified differences between lamprey and gnathostomes. The evolutionary significance of the conservation and differences is unclear. Thus, aside from the experimental depth, the core conclusions are not particularly novel. A revised discussion should emphasize the new insights the results provide into vertebrate evolution, cis-regulation, cis-regulatory evolution development, etc.

Reviewer #2:

Remarks to the Author:

The authors undertake a multi-pronged genomic study of neural crest gene regulation in the sea lamprey in order to compare a more ancestral version of the neural crest gene regulatory network (GRN) with previous studies in both early and later vertebrates. They use both RNA-seq and ATAC-seq at multiple time points on isolated dorsal neural tube and head tissue, followed by standard-type clustering, GO-term, and putative binding site analysis. Several putative enhancer sequences are successfully validated by transgenic reporter gene analysis, and a SoxE1 enhancer is tested cross-species in transgenic zebrafish, demonstrating ancient conservation of function.

Overall, I found this to be a solid, well-constructed study that makes useful contributions to the study of the evolution of development of the neural crest. The manuscript is well-written and the figures are clear. There are no major issues with the data, although I have some minor quibbles with some of the conclusions:

1. On lines 110-112 the authors write "our dataset captured the gene expression dynamics...while also providing insight into novel pathways that may be specific to lamprey neural crest development." I don't find this statement completely convincing. Although it is possible, it is also possible that these data merely represent noise in the analysis, perhaps through imperfect tissue isolation or imperfect genome annotation. Alternatively, they could represent a form of developmental systems drift whereby some of these genes could have been coopted into the neural crest pathway as part of the existing pathways rather than as novel participating pathways. The data as developed so far are not really sufficient to make these distinctions.

2. I wish that more than one enhancer had been tested in the zebrafish assay! While the successful result indicates that some degree of deep conservation exists, the limited data don't allow for any sort of assessment of to what extent. Is it just for one or two really essential regulators such as SoxE1? Or is there extensive deep conservation? The concept of deep conservation of this sort is no longer very novel in the sense that we know it can exist—the interesting questions now are more along the lines of to what extent, and in what developmental systems?

3. The binding site analysis for the SoxE1 enhancer is not very compelling absent some sort of controls suggesting how likely it would be to find some of these binding sites anyhow. This is a well-known issue with this sort of analysis, as binding sites are easy to find in the genome,

especially if no particular constraint is being placed on the exact assortment being looked for or how they should be ordered or spaced. Some simple controls using randomly selected non-coding sequences would help. A nice approach to this problem was taken by Kazemian et al (PMID 25173756), who considered conservation of specific sites among more closely-related species; that sort of approach might be possible here, for instance by using the two lamprey species. It's also not clear if a related enhancer is known in zebrafish, which would present another interesting point of comparison.

4. Along these lines, I take exception to the text in line 309-310 which states that "the lamprey SoxE1 enhancer harbours conserved binding site motifs..." The enhancer sequence is functionally conserved in zebrafish, but actual conservation of any binding sites has not been demonstrated. Similarly, the final conclusion that "the evolution of a combination of key transcription factor binding site motifs was central to neural crest GRN evolution. Conservation of these short motif sequences...is sufficient..." is really only quite loosely inferred by the data in hand. It's a reasonable hypothesis, but not yet strongly supported by the data.

Reviewer #3:

Remarks to the Author:

This paper describes a transcriptomic profiling of lamprey neural crest cells at different stages of its development. This basically extends previous similar efforts performed by the same group but now applying high throughput techniques. In addition, the authors use ATAC-seq to profile open regions in the genome to identify elements regulating expression of genes involved in neural crest biology. They went on to show that two of those regions are active in lamprey neural crest cells using a transgenic reporter approach. They also showed that one is also active in zebrafish embryos arguing for functional conservation across evolution. In my opinion, this paper is too descriptive with conclusions mainly based on bioinformatic classification of genes, lacking essential confirmation experiments. I also have some problems with various aspects of their experimental approach and with the accuracy of some of their statements. Some of these are specified below. In my opinion, although the data in this paper might be valuable, I think that it is too preliminary to be published in Nature Communications.

1- The authors dissected the dorsal neural tubes from embryos at stages T18, T20 and T21 to obtain the tissues that they consider to represent premigratory, migrating and postmigratory neural crest cells. This approach, however, does not result in the analysis of neural crest cells, as it clearly contains many other cell types (like those of the dorsal neural tube itself) that also change their characteristics as development proceeds. Therefore, I do not think that the resulting RNA-seq datasets can be evaluated as representing different stages of neural crest development. In addition, it sounds strange to use the neural tube of T21 embryos to sample postmigratory neural crest, as by definition migratory neural crest cells should have already left the neural tube, which would instead contain neural crest cells migrating later in development.

2- The authors discuss their RNA-seq datasets on the basis of a subset of the differentially expressed transcripts, which they select as representative. There are however some aspects of their interpretation that I find at least arguable. The following are just a few examples to illustrate my point.

2-A- A large number of the genes that seemed to be highly expressed in the "pre-migratory" set fell in GO classifications involving kidney, mesonephros, metanephros or inner ear (just to mention a few), which are hard to fit with neural crest. Importantly, they seem to be even more represented than those that fell into neural crest related classifications. Of course, this can be interpreted as new potential candidates for neural crest development, and in this case they should try to validate this interpretation with direct experimental data. However, following the same rationale of using expression of known neural crest genes to validate their approach, the odd-fitting data could as well be used as a counterargument to question their experimental approach (like the tissues used)

or the validity of the datasets or of the resulting conclusions.

2-B- the authors include genes like Wnt1, Pax7 or SoxE1 as part of the “neural crest specification module”. However genes like Wnt1 and Pax7, while they are indeed present in the dorsal neural tube with a specific expression dynamics during development (and therefore it is no surprise that they are present in the datasets) it is hard to link them to neural crest production according to published functional data. Actually, in the case of Pax genes Pax3 would be the right neural crest candidate, which I did not find in the datasets. SoxE1 plays no role in neural crest specification in lampreys but in its differentiation into cartilage (this is in McCauley and Bronner-Fraser (2006), *Nature* 441:750–752, which is actually not even cited in the present manuscript).

2-C- There are other genes like Pax6a or Col6a1, which according to Figure 1c are among the most strongly differentially upregulated in the T21 dataset that according to published information also play no role in the neural crest: Pax6a would actually fit with different stages of neural tube differentiation and Col6a1 is involved in striatal muscle differentiation. Why are they sort of excluded?

2-D. Some data is discussed using arguments to support their fitting with the expectations when the argument actually seems to go against it. For instance, they state “several Wnt homologues (Wnt5a/b, Wnt7a, Wnt8a) were depleted at T21 [postmigratory], consistent with studies showing that Wnt expression is modulated during neural crest delamination and migration with a switch from canonical Wnt signalling critical for specification to involvement of Wnt/PCP pathway during cell migration”. According to this, would not be expected to find upregulation of Wnt5a and downregulation of Wnt1 from T18 to T21 instead of the opposite, which is what the datasets show?

3- The authors show reporter expression for an enhancer within the SoxE1 genomic region compatible with activity in neural crest-containing tissues. However, at least superficially, the patterns observed in the embryos presented in the figure do not reproduce the expression of the endogenous SoxE1 in several ways. First, SoxE1 is only expressed in the branchial arches posterior to the mandibular, but enhancer activity seems to be extended further anteriorly, including the whole branchial and oral areas. Second, SoxE1 expression in the branchial area is restricted to the portion of the arches containing the cartilage, whereas the reporter seems to be expressed throughout the arches, at least in the whole mount images. The same seems to be true for the zebrafish experiments. This is important for several reasons, including the evaluation of their conclusion about the conservation of regulatory elements between zebrafish and lamprey but not with amphioxus. They use their reporter data to argue that the lamprey regulatory activity can be recognized by gnathostome transcription machinery, which they say that contrasts with the case of amphioxus, “where integration of the entire amphioxus SoxE locus and flanking genes into the zebrafish genome resulted in reporter expression in the developing neural tube and tail bud, but not in the neural crest”. Clearly, lamprey SoxE1 expression requires interactions with additional elements, and, therefore the proper experiment would be to use the whole SoxE1 genomic area in the reporter experiments (at least in zebrafish) for a proper lamprey/amphioxus comparison.

4- The authors speculate with the importance of the binding sites for transcription factors identified within the sequence of the SoxE1 enhancer using in silico methods. However, if they should show some experimental evidence for their functional relevance (way too often in vivo data does not follow in silico predictions), like making reporter analyses using mutant forms of the enhancer.

5- It is not clear to me why the Hoxa2-related data was included in this paper, as its relevance to the neural crest is not obvious. The authors describe potential new transcripts, including a lncRNA from this locus, apparently identified only based on annotation of RNA-seq data (at least this is what I could understand from the data provided). The existence of such transcripts and their expression in a neural crest-consistent pattern should be validated by independent methods. This is particularly important considering that the additional data used to somehow support relevance of these transcripts for neural crest development was a reporter analysis of a potential enhancer that seems active in the neural tube with no apparent contribution to the neural crest.

Reviewer #4:

Remarks to the Author:

In this study, the authors revisit the conservation of the gene regulatory network between jaw and jawless vertebrates by analyzing lamprey transcriptional profiles. Although the authors have been investigating this conservation for over 10 years, this is the most in-depth analysis performed to date. The authors present RNA-seq and ATAC-seq data from lamprey as well as testing activity of lamprey enhancer elements in higher vertebrate. The study is interesting and technical, however I have several significant concerns that reduce my enthusiasm for publication consideration of this manuscript in its current form. Specifically, this is primarily due to the analysis of a mixed population of cells that include pre-migratory, migratory and post-migratory neural crest as well as other cells types in the isolated tissue. Because pure cell populations were not isolated, it is inaccurate to state many of the conclusions that the authors make. Further, there is no second method to validate their RNA-seq results, insufficient comparisons between temporal stages to suggest trends in data or justification of dynamic analyses, and lack of statistical analyses that would confirm significant differences claimed in gene expression changes. Lastly, although the analysis and enhancer testing is of high quality, the conclusions made from these data are an oversimplification of the complexity of the multiple cell types analyzed. Therefore, I suggest these issues need to be addressed before further consideration.

Major concerns:

- 1) Throughout the manuscript the authors discuss the data as being specific for premigratory, migratory or post-migratory neural crest cells, however, this is incorrect. Pure populations of neural crest cells were not used in these experiments. Rather, dorsal neural tube or even whole head were harvested and used and not consistently compared. Therefore, to discuss this work as being specific for neural crest is inaccurate and oversimplifying the different tissue types that may have added to the results. This is inaccurate and needs to be justified.
- 2) The authors provide no validation of their RNA-seq results by a second method. Given the uncharacterized, heterogeneous populations analyzed, the authors should provide proof that the novel genes described are indeed expressed by neural crest cells.
- 3) Most of the transcriptomic results confirm previous work. While this helps to validate the approach, more focus should be given to the novel factors identified and presented.
- 4) The authors present a dynamic analysis of their RNA-seq data, however at times present only two time point comparisons rather than a comparison of all three (T18, T20, T21). That is, there is RNA-seq data at three successive time points but no appropriately details of the results to make this a 'dynamic' analysis. As an example, Fig 1c should include T18 vs T20 as well as T20 vs T21 comparisons in addition to the T18 vs T21 provided. Corresponding descriptions of the results should be included in the text.
- 5) The presentation of trends in data are not justified unless the authors can provide a statistical analysis of differential expression (for example in Fig. 1d). For example, is MEOX2 statistically different between any two timepoints in Fig 1di? If not, what is its relevance?
- 6) Clusters from WGCNA are based on similar changes in gene expression. Although interesting and informative, these clusters do not represent genes that necessarily function together or even genes that are expressed in the same cell types. The authors would have to either isolate pure populations of neural crest for sequencing or use in situ hybridizations for genes in each cluster to prove that they are neural crest specific.
- 7) Complete lists of differentially expressed genes should be provided as supplemental information. It is insufficient to perform such broad analyses and only report a handful of validating and novel genes as in Fig 1.
- 8) Please clarify what is meant to be represented by the T23 (whole head?) ATAC-seq sample. Are these additional post-migratory neural crest, and if so why weren't they included in the RNAseq analysis? Or is T23 meant to represent down-regulation of EMT genes for the T20 vs T23 comparison?
- 9) The authors state that "To focus our analyses on peaks associated with neural crest GRN genes,

consensus peaksets...were filtered to only contain peaks that were associated with genes enriched at T21." Why are only T21 enriched genes associated with neural crest GRN genes? If this is the case, why were T20 & T23 analyzed?

Minor concerns:

- 1) The scaling of Fig 1b, where 90% of the variance is represented on the smaller x axis, is misleading.
- 2) Please clarify "GO analysis of gene enriched at T20 & T21". Does this encompass genes enriched at both time points individually, or both timepoints as a combined sample. Where is the GO analysis of genes conversely reduced in T20 & T21 as in Fig 1di?
- 3) The authors describe RNA-seq and ATAC-seq quality control testing in the Methods sections, but fail to actually report the results within the manuscript.
- 4) The n values of successful reporter expression seem very low. What is the control experiment? If these experiments were repeated with a random element, how often would a similar pattern be seen?
- 5) Why was reporter expression weak in F0 generation?

We thank the reviewers for their helpful and constructive comments.

Reviewer #1 (Remarks to the Author):

The report describes the use of RNA-seq and ATAC-seq to identify genes and cis-regulatory elements involved in neural crest development in lampreys. Several putative cis regulatory elements are then validated by testing in lamprey and zebrafish. The work is technically impressive and well-executed. Novel features of the work are the choice of model systems, the depth of analysis, and the functional testing of putative elements. The results will serve as a comprehensive resource for neural crest biologists, and those interested in gene regulatory networks and their evolution. My only concern is that impact outside of the neural crest field may be limited.

The work supports conservation of the gene network driving neural crest development, and confirms a few previously identified differences between lamprey and gnathostomes. Thus, aside from the experimental depth, the core conclusions are not particularly novel. A revised discussion should emphasize the new insights the results provide into vertebrate evolution, cis-regulation, cis-regulatory evolution development, etc.

- We have revised the discussion to emphasise the fact that that this study provides functional support for the hypothesis that tissue-specific cis-regulatory functionality is independent of sequence constraint (lines 576-590).

Reviewer #2 (Remarks to the Author):

The authors undertake a multi-pronged genomic study of neural crest gene regulation in the sea lamprey in order to compare a more ancestral version of the neural crest gene regulatory network (GRN) with previous studies in both early and later vertebrates. They use both RNA-seq and ATAC-seq at multiple time points on isolated dorsal neural tube and head tissue, followed by standard-type clustering, GO-term, and putative binding site analysis. Several putative enhancer sequences are successfully validated by transgenic reporter gene analysis, and a SoxE1 enhancer is tested cross-species in transgenic zebrafish, demonstrating ancient conservation of function.

Overall, I found this to be a solid, well-constructed study that makes useful contributions to the study of the evolution of development of the neural crest. The manuscript is well-written and the figures are clear. There are no major issues with the data, although I have some minor quibbles with some of the conclusions:

1. On lines 110-112 the authors write “our dataset captured the gene expression dynamics...while also providing insight into novel pathways that may be specific to lamprey neural crest development.” I don’t find this statement completely convincing. Although it is possible, it is also possible that these data merely represent noise in the analysis, perhaps through imperfect tissue isolation or imperfect genome annotation. Alternatively, they could represent a form of developmental systems drift whereby some of these genes could have been coopted into the neural crest pathway as part of the existing pathways rather than as novel participating pathways. The data as developed so far are not really sufficient to make these distinctions.

- To address the reviewer’s concern that the results may represent noise in our analysis, we have used *in situ* hybridisation in lamprey embryos to confirm gene expression in the neural crest for genes that have not previously be implicated in early neural crest development, including *Zfhx3*, *Sdk1* and *Vitrin* (see Fig. 1g; Supplementary Fig. 2d; lines 215-240). Whether these genes are functionally pleiotropic and have been co-opted into the neural crest as part of existing pathways is an interesting question but is beyond the scope of this study.

2. I wish that more than one enhancer had been tested in the zebrafish assay! While the successful result indicates that some degree of deep conservation exists, the limited data don’t allow for any sort of assessment of to what extent. Is it just for one or two really essential regulators such as SoxE1? Or is there extensive deep conservation?

- Testing enhancers in the heterospecific assays was intended as a validation example, and not the main part of our study. We chose to focus our efforts on the lamprey SoxE1 enhancer and have now deepened the analysis by extending the conservation validation to an amniote model as well as by performing the functional upstream TF core analysis as suggested by the reviewer (see comments addressed below).

The concept of deep conservation of this sort is no longer very novel in the sense that we know it can exist—the interesting questions now are more along the lines of to what extent, and in what developmental systems?

- We thank the reviewer for this question/suggestion. We now also show that lamprey SoxE1 enhancer is also active in the chicken neural crest after electroporation (see Fig. 5b; lines 435-442). While deep conservation of enhancer activity has been

shown within the gnathostomes, it has never before been shown to exist from agnathans to gnathostomes therefore this is a novel finding (though some gnathostome enhancers have been previously shown to be active in agnathans).

3. *The binding site analysis for the SoxE1 enhancer is not very compelling absent some sort of controls suggesting how likely it would be to find some of these binding sites anyhow. This is a well-known issue with this sort of analysis, as binding sites are easy to find in the genome, especially if no particular constraint is being placed on the exact assortment being looked for or how they should be ordered or spaced. Some simple controls using randomly selected non-coding sequences would help.*

- The reviewer makes an excellent point. As suggested, we have now redone our genome-wide binding site motif analyses to identify all statistically significant enrichment in transcription site binding motifs in putative intergenic and intronic regulatory elements genome-wide using a random set of equivalent size of non-coding genomic regions as control. (see Fig. 2g,h,j; lines 753-756). We performed a similar analysis to identify enriched binding motifs in the SoxE1 enhancer (see Supplementary Fig. 7; lines 448-454).

A nice approach to this problem was taken by Kazemian et al (PMID 25173756), who considered conservation of specific sites among more closely-related species; that sort of approach might be possible here, for instance by using the two lamprey species.

- We thank the reviewer for this suggestion and we have now sequenced and analysed the SoxE1 enhancer region in the brook lamprey, which shows that the majority of the putative enriched transcription factor binding sites are conserved across species (see Supplementary Fig. 7; lines 454-461).

It's also not clear if a related enhancer is known in zebrafish, which would present another interesting point of comparison.

- Due to the lack of non-coding sequence conservation and the fact that synteny relationships are not well established between lampreys and zebrafish it is unfortunately not possible at this time to determine which zebrafish SoxE enhancer would be an equivalent to the lamprey enhancer.

4. *Along these lines, I take exception to the text in line 309-310 which states that “the lamprey SoxE1 enhancer harbours conserved binding site motifs...” The enhancer sequence is functionally conserved in zebrafish, but actual conservation of any binding sites has not been demonstrated.*

- We agree our use of language, which implies that we have shown the binding sites are conserved between lamprey and zebrafish, was misleading and apologise for this claim. We have now changed this sentence to: “Indeed, we show that the lamprey SoxE1 enhancer harbours putative canonical binding site motifs for several important neural crest transcription factors...” (see lines 566-569). In addition, as noted above, we have sequenced and analysed the SoxE1 enhancer region in the brook lamprey, which shows that the majority of the putative transcription factor binding sites are conserved across lamprey species (see Supplementary Fig. 7; lines 454-461).

Similarly, the final conclusion that “the evolution of a combination of key transcription factor binding site motifs was central to neural crest GRN evolution. Conservation of these short motif

sequences...is sufficient...” is really only quite loosely inferred by the data in hand. It’s a reasonable hypothesis, but not yet strongly supported by the data.

- To address this comment, we used the CRISPR/Cas9 system to knockout the expression of selected transcription factors (*hoxa2b*, *hoxb2a*, *hoxb3a*, *tfap2a*, *sox10*) in zebrafish embryos resulting from crossing our transgenic line carrying the lamprey *SoxE1* enhancer upstream of EGFP to zebrafish in which the branchial arch mesenchyme cells express DsRed reporter. We used the DsRed signal to sort the cells, extracted RNA and conducted qPCR to detect changes in *SoxE1*-driven reporter EGFP expression (See Fig. 5d, lines 462-475). The results show a down-regulation of EGFP expression after transcription factor knockdown, in comparison to controls (See Fig. 5e). This result supports a role for the affected transcription factors in the activation of the lamprey enhancer in the zebrafish genetic background, and therefore provides evidence that the binding interaction between zebrafish transcription factors and the lamprey regulatory element may enable the conservation of enhancer activity in the neural crest across species. This supports our hypothesis that the co-evolution of TF/enhancer binding might have played a key role in neural crest evolution and conservation of such interactions is key to the conservation of the neural crest GRN. We have added this argument to the discussion (see lines 572-590). However, we agree that we do not have strong enough evidence to support the use of the term “sufficient” in our conclusion and have now altered the text to reflect this (see lines 574-575).

Reviewer #3 (Remarks to the Author):

This paper describes a transcriptomic profiling of lamprey neural crest cells at different stages of its development. This basically extends previous similar efforts performed by the same group but now applying high throughput techniques. In addition, the authors use ATAC-seq to profile open regions in the genome to identify elements regulating expression of genes involved in neural crest biology. They went on to show that two of those regions are active in lamprey neural crest cells using a transgenic reporter approach. They also showed that one is also active in zebrafish embryos arguing for functional conservation across evolution. In my opinion, this paper is too descriptive with conclusions mainly based on bioinformatic classification of genes, lacking essential confirmation experiments. I also have some problems with various aspects of their experimental approach and with the accuracy of some of their statements. Some of these are specified below. In my opinion, although the data in this paper might be valuable, I think that it is too preliminary to be published in Nature Communications.

1- The authors dissected the dorsal neural tubes from embryos at stages T18, T20 and T21 to obtain the tissues that they consider to represent premigratory, migrating and postmigratory neural crest cells. This approach, however, does not result in the analysis of neural crest cells, as it clearly contains many other cell types (like those of the dorsal neural tube itself) that also change their characteristics as development proceeds. Therefore, I do not think that the resulting RNA-seq datasets can be evaluated as representing different stages of neural crest development.

In addition, it sounds strange to use the neural tube of T21 embryos to sample postmigratory neural crest, as by definition migratory neural crest cells should have already left the neural tube, which would instead contain neural crest cells migrating later in development.

- We have now adjusted the description of the sampled tissue as follows: T18: pre-migratory; T20: early-delaminating, T21: late-delaminating; T23: post-migratory (see line 106) and have made it clear that the analysed tissue encompasses both the neural tube or head, and its associated neural crest tissue at premigratory, early-delaminating, late-delaminating or post-migratory stages of development (see lines 124-126; 256). In particular, the region that we dissected in the T21 embryos would include the late delaminating neural crest cells that were in close proximity to the neural tube. While we acknowledge that the sampled tissue does contain both neural tube and neural crest tissue, we believe that the transcription signature in our data correlates strongly with what is expected for the neural crest.

2- The authors discuss their RNA-seq datasets on the basis of a subset of the differentially expressed transcripts, which they select as representative. There are however some aspects of their interpretation that I find at least arguable. The following are just a few examples to illustrate my point.

2-A- A large number of the genes that seemed to be highly expressed in the “pre-migratory” set fell in GO classifications involving kidney, mesonephros, metanephros or inner ear (just to mention a few), which are hard to fit with neural crest. Importantly, they seem to be even more represented than those that fell into neural crest related classifications. Of course, this can be interpreted as new potential candidates for neural crest development, and in this case they should try to validate this interpretation with direct experimental data. However, following the same rational of using expression of known neural crest genes to validate their approach, the odd-fitting data could as well be used as a counterargument to question their experimental approach (like the tissues used) or the validity of the datasets or of the resulting conclusions.

- To address this comment, we have now further investigated which genes fell into the “non-neural crest” GO term categories. This revealed that the most highly

expressed genes under these terms are already known to be expressed in neural crest development as well, but are also involved in processes such as kidney development. We have adjusted the text accordingly to reflect this (see lines 210-213). We have also further refined the list of GO terms to remove redundancies and added in a comparison between the terms associated with T18 and T21 (see Fig. 1f; lines 196-213). To make space for this expanded analysis as well as our *in situ* hybridisation analyses in the main figure, we moved the molecular function GO analysis to Supplementary Fig. 2b.

2-B- the authors include genes like *Wnt1*, *Pax7* or *SoxE1* as part of the “neural crest specification module”. However genes like *Wnt1* and *Pax7*, while they are indeed present in the dorsal neural tube with a specific expression dynamics during development (and therefore it is no surprise that they are present in the datasets) it is hard to link them to neural crest production according to published functional data. Actually, in the case of *Pax* genes *Pax3* would be the right neural crest candidate, which I did not find in the datasets. *SoxE1* plays no role in neural crest specification in lampreys but in its differentiation into cartilage (this is in McCauley and Bronner-Fraser (2006), *Nature* 441:750–752, which is actually not even cited in the present manuscript).

- *Pax3* is not annotated in the current lamprey genome assembly and previous work (McCauley and Bronner, 2002, *Gene* 287:129-139) has suggested that there may only be one *Pax3/7* group gene in the lamprey, which is currently annotated as *Pax7*. We thank the reviewer for pointing out this issue and have changed this annotation to *Pax3/7* in the manuscript (lines 163 and 515; Fig.1).
- Morpholino knockdown of *SoxE1* in lamprey embryos has been shown to lead to loss of *FoxD-A* expression in the pre-migratory neural crest (Sauka-Spengler et al., 2008, *Dev. Cell* 13:405-420), indicating *SoxE1* is involved in neural crest specification in the lamprey and therefore does support a neural crest signal in our data. While the role of *Wnt1* has not yet been investigated in the lamprey neural crest, work in chick embryos has shown that *Wnt1* signalling plays a role in the activation of neural crest migration (García-Castro et al., 2002, *Science* 297:848-851), therefore indicating that *Wnt1* plays an important role in neural crest development. We have added the necessary references to the text to clarify these points (see line 163)

2-C- There are other genes like *Pax6a* or *Col6a1*, which according to Figure 1c are among the most strongly differentially upregulated in the T21 dataset that according to published information also play no role in the neural crest: *Pax6a* would actually fit with different stages of neural tube differentiation and *Col6a1* is involved in striatal muscle differentiation. Why are they sort of excluded?

- We thank the reviewer for this comment and we now acknowledge that some of the genes that are up-regulated in our dataset, including *Pax6a*, are involved in neural tube development (lines 141-142). We have also now analysed a subset of these genes by *in situ* hybridization, including *Col6A1*, to show that they are also expressed in association with the developing neural crest (see Fig. 1g; Supplementary Fig. 2d; lines 214-240)

2-D. Some data is discussed using arguments to support their fitting with the expectations when the argument actually seems to go against it. For instance, they state “several *Wnt* homologues (*Wnt5a/b*, *Wnt7a*, *Wnt8a*) were depleted at T21 [postmigratory], consistent with studies showing that *Wnt* expression is modulated during neural crest delamination and migration with a switch

from canonical Wnt signalling critical for specification to involvement of Wnt/PCP pathway during cell migration". According to this, would not be expected to find upregulation of Wnt5a and downregulation of Wnt1 from T18 to T21 instead of the opposite, which is what the datasets show?

- Point taken. We have now edited the text to remove the statement "with a switch from canonical Wnt signalling critical for specification to involvement of Wnt/PCP pathway during cell migration", as we agree that our data do not conclusively support the switch from canonical to Wnt/PCP pathway gene expression (see line 141). Our data does however show *Wnt* gene expression is being dynamically modified over the time period we examined.

3- The authors show reporter expression for an enhancer within the SoxE1 genomic region compatible with activity in neural crest-containing tissues. However, at least superficially, the patterns observed in the embryos presented in the figure do not reproduce the expression of the endogenous SoxE1 in several ways. First, SoxE1 is only expressed in the branchial arches posterior to the mandibular, but enhancer activity seems to be extended further anteriorly, including the whole branchial and oral areas. Second, SoxE1 expression in the branchial area is restricted to the portion of the arches containing the cartilage, whereas the reporter seems to be expressed throughout the arches, at least in the whole mount images.

- Again, we thank the reviewer for this comment and the opportunity to clarify our data. We have used hybridisation chain reaction (HCR) *in situ* experiments to show an overlap of SoxE1 enhancer reporter expression and endogenous SoxE1 expression in the delaminating neural crest at T23 and as well as at T26 in the branchial arch cartilage (see Fig. 3b xii-xiii; lines 353-356). In addition, we now present sections of the T26 embryo (Fig. 3b xi) clearly indicating that SoxE1 enhancer reporter expression is localised to the branchial arch cartilage, as well as the cranial ganglia, but is not found throughout the arches, as suggested by the reviewer.

The same seems to be true for the zebrafish experiments.

- To address the reviewer's concern, we have now used immunostaining and sectioning of transgenic zebrafish embryos carrying the SoxE1 enhancer upstream of GFP to show that reporter expression is restricted to neural crest derivatives in the head, including the branchial arch mesenchyme, and is excluded from the ectoderm (see Supplementary Fig. 6; lines 430-434).

This is important for several reasons, including the evaluation of their conclusion about the conservation of regulatory elements between zebrafish and lamprey but not with amphioxus. They use their reporter data to argue that the lamprey regulatory activity can be recognized by gnathostome transcription machinery, which they say that contrasts with the case of amphioxus, "where integration of the entire amphioxus SoxE locus and flanking genes into the zebrafish genome resulted in reporter expression in the developing neural tube and tail bud, but not in the neural crest". Clearly, lamprey SoxE1 expression requires interactions with additional elements, and, therefore the proper experiment would be to use the whole SoxE1 genomic area in the reporter experiments (at least in zebrafish) for a proper lamprey/amphioxus comparison.

- We apologise if our description has misguided the reviewer into thinking that our study was a direct comparison with the amphioxus experiment. While we agree that the experiment proposed by the reviewer would be very interesting, we believe that it lies outside of the scope of this study. We have now discussed our findings in light

of other experiments, such as the described one, but have tried our utmost to clarify our statement and render the language more precise (see lines 555-562)

4- *The authors speculate with the importance of the binding sites for transcription factors identified within the sequence of the SoxE1 enhancer using in silico methods. However, if they should show some experimental evidence for their functional relevance (way too often in vivo data does not follow in silico predictions), like making reporter analyses using mutant forms of the enhancer.*

- To address this comment, we used the CRISPR/Cas9 system to knockout the expression of selected transcription factors (*hoxa2b*, *hoxb2a*, *hoxb3a*, *tfap2a*, *sox10*) in zebrafish embryos resulting from crossing our transgenic line carrying the lamprey *SoxE1* enhancer upstream of EGFP to zebrafish in which the branchial arch mesenchyme cells express DsRed reporter. We used the DsRed signal to sort the cells, extracted RNA and conducted qPCR to detect changes in SoxE1-driven reporter EGFP expression (See Fig. 5d, lines 462-475). The results show a down-regulation of EGFP expression after transcription factor knockdown, in comparison to controls (See Fig. 5e). This result supports a role for the affected transcription factors in the activation of the lamprey enhancer in the zebrafish genetic background, and therefore provides evidence that the binding interaction between zebrafish transcription factors/lamprey regulatory element may enable the conservation of enhancer activity in the neural crest across species.

5- *It is not clear to me why the Hoxa2-related data was included in this paper, as its relevance to the neural crest is not obvious.*

- The *Hoxa2*-related data was included to demonstrate that our ATAC-seq datasets can be used to further hone known enhancer regions. A recently published paper by Parker et al. 2019 in *Nature Communications* further refined the 4 kb *Hoxa2* neural crest enhancer in the lamprey to a short 1.5 kb region (elementA), which overlaps with an ATAC positive region in our dataset. Therefore, we have adjusted our analysis of the *Hoxa2* locus to emphasise that our ATAC-seq dataset can be used to identify and refine neural crest enhancer elements (See Fig. 4; lines 380-387).

The authors describe potential new transcripts, including a lncRNA from this locus, apparently identified only based on annotation of RNA-seq data (at least this is what I could understand from the data provided). The existence of such transcripts and their expression in a neural crest-consistent pattern should be validated by independent methods. This is particularly important considering that the additional data used to somehow support relevance of these transcripts for neural crest development was a reporter analysis of a potential enhancer that seems active in the neural tube with no apparent contribution to the neural crest.

- Point taken. We have now attenuated our statement and replaced the phrase “during neural crest development” with “in the neural tube and/or neural crest at T21.” (see lines 413-414).

Reviewer #4 (Remarks to the Author):

In this study, the authors revisit the conservation of the gene regulatory network between jaw and jawless vertebrates by analyzing lamprey transcriptional profiles. Although the authors have been investigating this conservation for over 10 years, this is the most in-depth analysis performed to date. The authors present RNA-seq and ATAC-seq data from lamprey as well as testing activity of lamprey enhancer elements in higher vertebrate. The study is interesting and technical, however I have several significant concerns that reduce my enthusiasm for publication consideration of this manuscript in its current form. Specifically, this is primarily due to the analysis of a mixed population of cells that include pre-migratory, migratory and post-migratory neural crest as well as other cells types in the isolated tissue. Because pure cell populations were not isolated, it is inaccurate to state many of the conclusions that the authors make. Further, there is no second method to validate their RNA-seq results, insufficient comparisons between temporal stages to suggest trends in data or justification of dynamic analyses, and lack of statistical analyses that would confirm significant differences claimed in gene expression changes. Lastly, although the analysis and enhancer testing is of high quality, the conclusions made from these data are an oversimplification of the complexity of the multiple cell types analyzed. Therefore, I suggest these issues need to be addressed before further consideration.

Major concerns:

1) *Throughout the manuscript the authors discuss the data as being specific for premigratory, migratory or post-migratory neural crest cells, however, this is incorrect. Pure populations of neural crest cells were not used in these experiments. Rather, dorsal neural tube or even whole head were harvested and used and not consistently compared. Therefore, to discuss this work as being specific for neural crest is inaccurate and oversimplifying the different tissue types that may have added to the results. This is inaccurate and needs to be justified.*

- We thank the reviewer for this remark and have now made it clear that the analysed tissue encompasses both the neural tube or head, and its associated neural crest tissue at premigratory, early-delaminating, late-delaminating or post-migratory stages of development (see lines 124-126; 256).

2) *The authors provide no validation of their RNA-seq results by a second method. Given the uncharacterized, heterogeneous populations analyzed, the authors should provide proof that the novel genes described are indeed expressed by neural crest cells.*

3) *Most of the transcriptomic results confirm previous work. While this helps to validate the approach, more focus should be given to the novel factors identified and presented.*

- Point taken. To provide a second method for the validation of our RNA-seq results and test some of the novel factors, we have used *in situ* hybridisation in lamprey embryos. We confirm gene expression in the neural crest for a number of genes that have not previously be implicated in early neural crest development, including *Zfhx3*, *Sdk1* and *Vitrin* (see Fig. 1g; Supplementary Fig. 2d; lines 214-240).

4) *The authors present a dynamic analysis of their RNA-seq data, however at times present only two time point comparisons rather than a comparison of all three (T18, T20, T21). That is, there is RNA-seq data at three successive time points but no appropriately details of the results to make this a 'dynamic' analysis. As an example, Fig 1c should include T18 vs T20 as well as T20 vs T21 comparisons in addition to the T18 vs T21 provided. Corresponding descriptions of the results should be included in the text.*

- As recommended by the reviewer's, we have now generated a volcano plot

representation of genes differentially expressed at T18 vs T20, as well as T20 vs T21 stages (see Supplementary Fig. 2a) and added a description to the manuscript (see lines 145-151).

5) *The presentation of trends in data are not justified unless the authors can provide a statistical analysis of differential expression (for example in Fig. 1d). For example, is MEOX2 statistically different between any two timepoints in Fig 1d? If not, what is its relevance?*

- We now also report the gene significance (GS; i.e. the correlation between the gene and the indicated trait [T18 or T21]), module membership (MM; i.e. the correlation of the module eigengene and the gene expression profile) and corresponding *p*-values (p.MM) for each gene in each WGCNA cluster (see Supplementary File 2). We no longer highlight *Meox2* in Fig. 1d as we do not discuss it in the text.

6) *Clusters from WGCNA are based on similar changes in gene expression. Although interesting and informative, these clusters do not represent genes that necessarily function together or even genes that are expressed in the same cell types. The authors would have to either isolate pure populations of neural crest for sequencing or use in situ hybridizations for genes in each cluster to prove that they are neural crest specific.*

- As stated above, we have used whole mount *in situ* hybridisation in lamprey embryos to confirm gene expression in the neural crest for selected genes (see Fig. 1g).

7) *Complete lists of differentially expressed genes should be provided as supplemental information. It is insufficient to perform such broad analyses and only report a handful of validating and novel genes as in Fig 1.*

- This information is now provided in Supplementary File 1.

8) *Please clarify what is meant to be represented by the T23 (whole head?) ATAC-seq sample. Are these additional post-migratory neural crest, and if so why weren't they included in the RNAseq analysis? Or is T23 meant to represent down-regulation of EMT genes for the T20 vs T23 comparison?*

- These represent late post-migratory but pre-differentiated NC. We expect that the enhancers that are involved in activating genes needed for migration would still be active in this tissue. Unfortunately, we did not collect the corresponding RNA-seq data.

9) *The authors state that "To focus our analyses on peaks associated with neural crest GRN genes, consensus peaksets...were filtered to only contain peaks that were associated with genes enriched at T21." Why are only T21 enriched genes associated with neural crest GRN genes? If this is the case, why were T20 & T23 analyzed?*

- To limit the number of peaks to be analysed and focus our study, we chose to hone onto the programme being activated during EMT by analysing the putative elements associated with genes that were significantly enriched at our latest stage of RNA-seq analysis (T21), when EMT is occurring. T20 and T23 were also analysed in order to detect global dynamic changes in chromatin accessibility over this period of development (and thus dynamically "employed" putative elements).

Minor concerns:

1) *The scaling of Fig 1b, where 90% of the variance is represented on the smaller x axis, is misleading.*

- The scaling of Fig. 1b has been adjusted so that the x-axis is longer than the y-axis.

2) *Please clarify “GO analysis of gene enriched at T20 & T21”. Does this encompass genes enriched at both time points individually, or both timepoints as a combined sample.*

- This encompasses genes enriched at each time point compared to T18. We have clarified this in the text (see line 201)

Where is the GO analysis of genes conversely reduced in T20 & T21 as in Fig 1di?

- We have now performed GO analysis of genes that are reduced at T21 in comparison to T18 i.e. enriched at T18 (see Fig. 1f; lines 196-200). As this analysis was very similar for T20 vs T18, we only show it for T21.

3) *The authors describe RNA-seq and ATAC-seq quality control testing in the Methods sections, but fail to actually report the results within the manuscript.*

- We include results of quality control testing in Supplementary Figures 1, 4 and 5 and refer to these figures in lines 122-124, 259 and 663.

4) *The n values of successful reporter expression seem very low. What is the control experiment? If these experiments were repeated with a random element, how often would a similar pattern be seen?*

- Several other elements were tested (28). Only the elements reported here showed these specific patterns of expression in the neural crest. The numbers of positive embryos depends largely on the transgenesis efficiency and the quality of the embryos.

5) *Why was reporter expression weak in F0 generation?*

- We suspect the enhancer is being shut down by epigenetic mechanisms in the zebrafish. However, investigation of this hypothesis is beyond the scope of this study.

Reviewers' Comments:

Reviewer #2:

Remarks to the Author:

The authors have performed a number of new experiments and analyses, and have made numerous modifications to the text. Overall, I am satisfied with these modifications and feel that my main objections have been addressed. However, there are two points on which I still feel like the analysis is oversold, and recommend that the authors tone down their interpretation to better suit the evidence.

One place is in the comparison between the two lamprey species (Supp. Fig. 7). I appreciate that the authors performed this analysis, but I think their conclusion over-reaches. There appears to be very little sequence divergence across the whole of the region that they compare. (It would be interesting to know if this is true generally, or just for this and other potential enhancer regions, but that's another thing.) Given that, I don't think it's reasonable to impute particular importance to the putative binding sites of interest, as they don't stand out as better conserved than any other small subsequence in this region.

The other place is Fig. 5d,e/S8, with the CRISPR knockdowns. This is a good experiment and confirms the larger point that this region is important for regulation of SoxE1. However, I don't see how one can conclude that "a combination" of sites is required from this experiment. The test was only done using a combination, not individual guide RNAs. As a result, one cannot differentiate between a combination of sites being required, and a single site being required with the remainder having no effect. I'm suspect that the authors' interpretation that combinatorial TF binding plays a role is correct, but I don't see that they provide compelling evidence to support it. This factors into both the description of the experiment (lines 472-476) and the Discussion (lines 573-575 and 576-579).

Reviewer #3:

Remarks to the Author:

This work provides a comprehensive catalog of genes belonging the gene regulatory network controlling different stages of neural crest development in lamprey by combining RNA-seq and ATAC-seq analyses of neural crest cells at different stages of its development. The authors also analyze the activity putative neural crest enhancers identified in this study by transgenic reporter experiments and address their evolutionary conservation.

In this revised version, the authors have successfully addressed my main concerns regarding experimental approach and interpretation of their data. However, I still think that the study remains at a descriptive level. The possible relevance of the genes found in the transcriptomic analyses is discussed on the basis of bioinformatic classification of genes and of known roles according to published genetic experiments performed in other vertebrate species. However, there is no effort towards validating the functional relevance of any of the factors that they identify as unique to lampreys, and therefore, it is not possible to evaluate the significance of these findings, which limits the impact of this work.

Reviewer #4:

Remarks to the Author:

It is clear that there is an extensive amount of effort that has gone into the preparation of this manuscript and its revision. The authors have now revised the results and discussion sections in an attempt to strengthen their core conclusions, but a couple of the main concerns remain. First, that functionality is independent of sequence constraints is novel but not unexpected and not particularly novel. The rework of the Discussion should also include mention of how the authors

interpret how unconstrained cis-regulation evolved, based on their findings. Second, in response to my concern that the sequenced samples are impure/heterogeneous (shared by Reviewer 3's main concern), the authors have amended their description of the samples and provided in situ hybridization data supporting expression in neural crest cells. However, the small number of in situ patterns shown (n=3) along with the correlation to expectation seems insufficient validation of complex sequencing data. A better approach would be to display strong correlation between a new pure neural crest population and an existing stage-matched sample. Again, this point appears to echo the primary concern of Reviewer 3; that is, renaming the sample does not change its heterogeneous composition even if it 'correlates with expectations'.

REVIEWERS' COMMENTS:

Reviewer #2 (Remarks to the Author):

The authors have performed a number of new experiments and analyses, and have made numerous modifications to the text. Overall, I am satisfied with these modifications and feel that my main objections have been addressed. However, there are two points on which I still feel like the analysis is oversold, and recommend that the authors tone down their interpretation to better suit the evidence.

One place is in the comparison between the two lamprey species (Supp. Fig. 7). I appreciate that the authors performed this analysis, but I think their conclusion over-reaches. There appears to be very little sequence divergence across the whole of the region that they compare. (It would be interesting to know if this is true generally, or just for this and other potential enhancer regions, but that's another thing.) Given that, I don't think it's reasonable to impute particular importance to the putative binding sites of interest, as they don't stand out as better conserved than any other small subsequence in this region.

We appreciate the reviewer's concern and have adjusted the text to tone down our interpretation (see lines 392-395)

The other place is Fig. 5d,e/S8, with the CRISPR knockdowns. This is a good experiment and confirms the larger point that this region is important for regulation of SoxE1. However, I don't see how one can conclude that "a combination" of sites is required from this experiment. The test was only done using a combination, not individual guide RNAs. As a result, one cannot differentiate between a combination of sites being required, and a single site being required with the remainder having no effect. I'm suspect that the authors' interpretation that combinatorial TF binding plays a role is correct, but I don't see that they provide compelling evidence to support it. This factors into both the description of the experiment (lines 472-476) and the Discussion (lines 573-575 and 576-579).

We have adjusted the text to address these concerns (see lines 405-406, 482-483)

Reviewer #3 (Remarks to the Author):

This work provides a comprehensive catalog of genes belonging the gene regulatory network controlling different stages of neural crest development in lamprey by combining RNA-seq and ATAC-seq analyses of neural crest cells at different stages of its development. The authors also analyze the activity putative neural crest enhancers identified in this study by transgenic reporter experiments and address their evolutionary conservation.

In this revised version, the authors have successfully addressed my main concerns regarding experimental approach and interpretation of their data. However, I still think that the study remains at a descriptive level. The possible relevance of the genes found in the transcriptomic analyses is discussed on the basis of bioinformatic classification of genes and of known roles according to published genetic experiments performed in other vertebrate species. However, there is no effort towards validating the functional relevance of any of the factors that they identify as unique to lampreys, and therefore, it is not possible to evaluate the significance of these findings, which limits the impact of this work.

We thank the reviewer for their comments. We believe that our transcriptomic analyses provide a rich resource for the neural crest community for identifying neural crest GRN players that are either unique to lampreys or shared with other vertebrates. Future studies in our lab (and we hope other labs) will continue to mine this data and deepen the functional analyses and mechanistic understanding.

Reviewer #4 (Remarks to the Author):

It is clear that there is an extensive amount of effort that has gone into the preparation of this manuscript and its revision. The authors have now revised the results and discussion

sections in an attempt to strengthen their core conclusions, but a couple of the main concerns remain. First, that functionality is independent of sequence constraints is novel but not unexpected and not particularly novel. The rework of the Discussion should also include mention of how the authors interpret how unconstrained cis-regulation evolved, based on their findings.

We have elaborated on our discussion of unconstrained *cis*-regulatory element evolution, including a discussion on the role that combinatorial transcription factor binding plays in facilitating positive selection of transcription factor binding site motifs (see lines 493-503).

Second, in response to my concern that the sequenced samples are impure/heterogeneous (shared by Reviewer 3's main concern), the authors have amended their description of the samples and provided in situ hybridization data supporting expression in neural crest cells. However, the small number of in situ patterns shown (n=3) along with the correlation to expectation seems insufficient validation of complex sequencing data.

We have validated the expression patterns of 9 genes from our transcriptome and believe this is sufficient for the first description of this recourse. Future studies will continue to mine this extensive resource and analyse the expression of further genes.

A better approach would be to display strong correlation between a new pure neural crest population and an existing stage-matched sample. Again, this point appears to echo the primary concern of Reviewer 3; that is, renaming the sample does not change its heterogeneous composition even if it 'correlates with expectations'.

We appreciate the reviewer's concern. Unfortunately, obtaining a pure population of premigratory neural crest cells from lamprey embryos is not possible at this time. Firstly we do not have an appropriate reporter that would allow us to fluorescently label the premigratory and early delaminating neural crest population that is under investigation in our study. The enhancers isolated in this study start driving fluorescence in the migrating neural crest and only in the specific subpopulation population within the branchial arches but do not encompass the entirety of the cranial neural crest. Second and more important, we have spent much time and continue trying to optimise FAC-sorting experiments in lamprey, and while we had some success with later larvae, obtaining purified cells from very early neurula remains challenging. The technical difficulties are linked to the nature of the embryonic lamprey cells, which are packed with voluminous vitamin D-rich yolk platelets that auto-fluoresce and cannot be easily distinguished from cells. Another, maybe even more critical concern, and a significant obstacle to the approach is the integrity of cells after dissociation. Lamprey embryonic cells are tightly packed and require enzymatic association. The cells are also susceptible to increase in temperature; however, a majority of enzymes used in cell dissociation are inactive at low temperatures. Dissociating then mostly relies on physical manipulation with depletion in Ca⁺⁺/Mg⁺⁺ ions – but this is not a very efficient and vast majority of cells are lysed during the process. Due to all these issues, we have found that FACS to be very unreliable in this model organism, at least for very early embryos under study here.

In response to reviewer's concerns, we have included a discussion of the limitations of the study as a result of analysing a dissected heterogeneous population of cells (see lines 506-514)